



# The importance of management information and soil moisture representation for simulating tillage effects on N₂O emissions in LPJmL5.0-tillage

Femke Lutz[1,2], Stephen DelGrosso[3], Stephen Ogle[4], Stephen Williams[4], Sara Minoli[1], Susanne Rolinski[1], Jens Heinke[1], Jetse J. Stoorvogel[2], and Christoph Müller[1]

[1]Potsdam Institute for Climate Impact Research (PIK), member of the Leibniz Association, P.O. Box 60 12 03, 14412 Potsdam, Germany
[2]Wageningen University, Soil Geography and Landscape Group, P.O. Box 47, 6700 AA Wageningen, the Netherlands
[3]USDA-ARS, Soil management and Sugar Beet Research Unit, 2150 Centre Ave. Bldg. D, Fort Collins, CO 80526, USA
[4]NREL, Colorado State University, Fort Collins, CO 80523, USA

**Correspondence:** Femke Lutz (femke.lutz@pik-potsdam.de)

**Abstract.** No-tillage is often suggested as a strategy to reduce greenhouse gas emissions. Modeling tillage effects on nitrous oxide (N₂O) emissions is challenging and subject to large uncertainties, as the processes producing the emissions are complex and strongly non-linear. Previous findings have shown deviations between the LPJmL5.0-tillage model and results from meta-analysis on global estimates of tillage effects on N₂O emissions. Here we tested LPJmL5.0-tillage at four different experimental

sites across Europe and the USA, to verify whether deviations in N₂O emissions under different tillage regimes result from a lack of detailed information on agricultural management and/or the representation of soil water dynamics. Model results were compared to observational data and outputs from field-scale DayCent simulations. DayCent has been successfully applied for the simulation of N₂O emissions and provides a richer data base for comparison than non-continuous measurements at the experimental sites. We found that adding information on agricultural management improved the simulation of tillage effects on

N₂O emissions in LPJmL. We also found that LPJmL overestimated N₂O emissions as well as the effects of no-tillage on N₂O emissions, whereas DayCent tended to underestimate the emissions of no-tillage treatments. LPJmL showed a general bias to over-estimate soil moisture content. Modifications of hydraulic properties in LPJmL in order to match properties assumed in DayCent, as well as of the parameters related to residue cover, improved the overall simulation of soil water as well as the N₂O emissions simulated under tillage and no-tillage separately. However, the effects of no-tillage (shifting from tillage to

no-tillage) did not improve. Advancing the current state of information on agricultural management as well as improvements in soil moisture highlight the potential to improve LPJmL5.0-tillage and global estimates of tillage effects on N₂O emissions.

## 1 Introduction

Agricultural fields are often tilled to suppress weeds, incorporate crop residues, aerate the soil, prepare the seedbed and improve infiltration. The resulting changes in physical and chemical properties of the soil affect several biochemical processes, including

the formation of greenhouse gases (GHG). Many field-scale models and experiments evaluated the effects of tillage and no-





tillage on GHG and soil organic carbon (SOC) (Álvaro-Fuentes et al., 2012; Del Grosso et al., 2009; Jin et al., 2017; Oorts et al., 2007). Nitrous oxide ($N_2O$) is a very strong GHG and predominantly emitted in agricultural production (Ciais et al., 2014; Smith, 2017). However, studies reported mixed results for the impacts of adapting no-tillage on $N_2O$ emissions from croplands (Deng et al., 2016; Venterea et al., 2011). For instance, no-tillage was found to increase $N_2O$ emissions (Mei et al.,

2018; Van Kessel et al., 2013), decrease $N_2O$ emissions (Deng et al., 2016; Plaza-Bonilla et al., 2018; Yoo et al., 2016) or having no significant effects (Alvarez et al., 2012; Boeckx et al., 2011) in comparison to conventional tillage systems.

Soils emit $N_2O$ through a series of processes involving denitrification and nitrification. These processes are driven by microbial activity and strongly respond to soil properties such as moisture, temperature, oxygen, mineral N, and organic carbon (Mosquera et al., 2005; Snyder et al., 2009; Van Kessel et al., 2013). These soil properties are affected by tillage (Lutz

et al., 2019a, c) and other management practices (e.g., fertilizer application and residue treatment) (Van Kessel et al., 2013). Due to the complexity of the system, the simulation of tillage effects on $N_2O$ emissions is challenging and subject to large uncertainties.

Lutz et al. (2019a) extended a dynamic global vegetation, hydrology and crop model to explicitly account for the effects of tillage in the simulations of biogeochemical cycles, hydrology and crop yields. This enables simulations of the effects of

tillage on crop productivity, the water, carbon and nitrogen cycles, including $N_2O$ emissions at the global scale. However, they found that simulated $N_2O$ emissions from no-tillage exceeded values in most of the climate zones reported in meta-analyses. These deviations between observations and simulations of tillage effects on $N_2O$ emissions can have several different causes, including missing processes and lack of process understanding. Also the parameterization of implemented processes as well as detailed information on management aspects that are explicitly addressed in the model can lead to model deficiencies that

could cause the mismatch between observations and simulations.

For example, as detailed information about agricultural management practices is lacking for global-scale applications, assumptions on agricultural management are necessary in these global simulations about e.g., the type, amount and timing of fertilizer applications. Detailed information on fertilization can typically be dealt with in field-scale modeling experiments, whereas at the global scale, there is only general information on fertilization (e.g. Mueller et al., 2012; Potter et al., 2010)

which is characterized by gaps and uncertainties (Erb et al., 2017). These generalizations may be a significant contributor to the overall uncertainty for agricultural impact assessments. For instance, Folberth et al. (2019) found that differences in management assumptions (about e.g., growing season, and fertilization) resulted in substantial differences in modeled crop yields using the same crop model.

Second, the formation of $N_2O$ in soils is very sensitive to soil moisture (Butterbach-Bahl et al., 2013). How the effect of

tillage on soil moisture is simulated is thus another source of uncertainty that could explain the inaccuracy in modeling tillage effects on $N_2O$ emissions.

In this study, we test the importance of management information as well as the representation of soil water dynamics for the ability to simulate $N_2O$ emissions under different tillage regimes with LPJmL5.0-tillage (Lutz et al., 2019a), for four different experimental sites across Europe and the USA. Simulation results are compared to measurements of $N_2O$ emissions

from experimental studies under tillage and no-tillage in different simulation experiments, varying from using observed site-

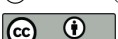



specific information to using the default assumptions usually applied in the global-scale simulations. Because of the importance of soil moisture for $N_2O$ emissions, we test the accuracy of the simulated soil moisture dynamics and its effects on $N_2O$ emissions against observations. As simulating tillage effects on $N_2O$ emissions is generally challenging, we use the site-specific model DayCent (Del Grosso et al., 2009; Parton et al., 1996), which has previously been applied at the study sites

as a benchmark and to provide more detailed information on soil hydrology than the sparse observations. DayCent is a well-established model that has been used for questions related to agricultural impact assessments at various scales (e.g. Begum et al., 2019; Del Grosso et al., 2009; Del Grosso et al., 2002; Gryze et al., 2010). DayCent can be used as a benchmark for which the underlying mechanisms can be analyzed and used for improvements of LPJmL5.0-tillage, even though the performance of DayCent has to be compared to observations first.

## 2    Material and methods

### 2.1    Overview

In Lutz et al. (2019a), model results deviated from meta-analyses when comparing simulated tillage effects on $N_2O$ emissions. First, we tested whether the deviations are due to a lack of detailed management information. Four experimental sites for which detailed information on management are available were identified. On those sites, LPJmL5.0-tillage was run using

management assumptions usually used in a global simulation experiment (LPJmL.G.Orig). To find out if LPJmL5.0-tillage performed better with detailed information on management, we also applied LPJmL5.0-tillage using detailed site-specific management information to derive inputs (LPJmL.D.Orig).

        The site-specific DayCent model was used as benchmark to analyze the underlying mechanisms of the $N_2O$ producing processes. For all the simulations of DayCent, detailed information of management was used. Except for the experimental

site in Boigneville, DayCent has been used and calibrated for field-scale assessments at the chosen sites (i.e. Campbell et al., 2014; Del Grosso et al., 2009; Yang et al., 2017). Therefore, we expect it to perform better on simulating the effects of tillage on $N_2O$ emissions than LPJmL. We also expect to learn from the underlying mechanisms simulated by DayCent and to use this information for improving process representation and parameterization in LPJmL. All model versions considered here require similar inputs (soil properties, vegetation type, land management information, latitude, daily precipitation, and daily

air temperature (minimum and maximum).

### 2.2    LPJmL5.0-tillage

LPJmL5.0-tillage is a dynamic global vegetation, hydrology and crop model that simulates nitrogen (N), carbon (C) and water dynamics in natural and agricultural ecosystems. Soils are represented by five hydrologically active layers, with different layer thicknesses.

LPJmL5.0-tillage (in the following referred to as LPJmL) uses three litter pools; representing surface litter, incorporated litter and below-ground litter as well as two soil organic matter (SOM) pools per soil layer, which are characterized by fast





and slow decomposition rates, respectively, and by separate C and N components for each pool. The surface litter pool consists of crop residues which are not removed at harvest or incorporated into the first soil layer through tillage. Residue cover is calculated from the surface litter following Gregory (1982). This residue cover intercepts some rainfall, promotes infiltration

into the soil, and limits soil evaporation. Moreover, the presence of a residue cover insulates the soil from air temperature fluctuations. The effects of residue cover on soil water dynamics and soil temperature fluctuations are thoroughly described in Lutz et al. (2019a).

Surface litter decomposes and is incorporated through bioturbation and tillage, forming the incorporated litter pool in the first layer. The below ground litter pool includes crop roots that remain in the soil after harvest. All pools are subjected

to decomposition, which is driven by the moisture content and temperature of the soil (for the incorporated litter and below-ground litter pool), or of the moisture content and temperature of the surface litter (surface litter pool). A fixed fraction of the decomposed litter is mineralized and emitted as $CO_2$, whereas the remaining C is transferred to the SOM pool, where it is then subject to soil C decomposition (see also Von Bloh et al., 2018). The mineralized N is added to the $NH_4^+$ pool which is subject to further transformations into other forms of nitrogen (Von Bloh et al., 2018).

Nitrification and denitrification are simulated throughout the entire soil profile and are dependent on the water-filled pore space (WFPS), soil temperature, $NH_4^+$, pH, SOC (denitrification) and $NO_3^-$. The $N_2O$ emissions from denitrification increases exponentially when the WFPS reaches a threshold value of $\geq 90\%$, as denitrification occurs only in oxygen deficit conditions (see also Krysanova and Wechsung, 2000).

In addition to tillage effects on residues (i.e. incorporating residues into the soil), tillage affects the hydraulic properties

of the soil by decreasing the bulk density. Soil hydraulic parameters are calculated through a pedotransfer function (PTF) from Saxton and Rawls (2006) which uses soil texture, SOM, and bulk density changes to calculate field capacity (FC), wilting point (WP), saturation (WSAT) and the saturated hydraulic conductivity (Ksat). The hydraulic parameters determine the water holding capacity- and the water dynamics of the soil. For instance, soil water above WSAT runs off as lateral runoff, while remaining soil water above FC percolates to the next soil layer and generates lateral subsurface runoff or vertical seepage from

the soil column.

A full overview of the tillage implementation into LPJmL5.0 as well as affected soil properties and processes can be found in Lutz et al. (2019a), the nitrogen implementation is described by Von Bloh et al. (2018) and a comprehensive description of the LPJmL model is provided by Schaphoff et al. (2018) and Schlüter et al. (2018).

## 2.3 DayCent

The DayCent ecosystem model simulates crop growth, soil water, C and nutrient dynamics (N, P) in natural and agricultural ecosystems (Del Grosso et al., 2009; Parton et al., 1998). The soil is represented by user-specified layers which are hydrologically active. DayCent has two litter pools, representing surface-litter and below-ground litter and three SOM pools (active, slow and passive) characterized by different decomposition rates.

The active and the slow organic matter pools have surface as well as soil components while the passive pool has only

a soil component. The litter pools are partitioned into structural and metabolic pools as a function of the lignin to N ratio in





the residue, which are subject to decomposition. Decomposition products of litter supply the SOM pools (surface active, soil active, surface slow and soil slow) and are partitioned among pools based on lignin content. Decomposition of litter and soil organic matter and nutrient mineralization are a function of substrate availability, substrate quality (lignin content, C:N ratio), soil moisture, soil temperature and tillage intensity. N-mineralization, N-fertilization and N-fixation supply the N-pools. $NO_3^-$

is distributed throughout the soil profile, whereas $NH_4^+$ is confined to the top 10 cm. $NO_3^-$ and $NH_4^+$ can then be taken up by plants, leached to lower layers ($NO_3^-$ only) or transformed to N gas emissions (e.g. $N_2O$) through nitrification or denitrification (Del Grosso et al., 2000; Parton et al., 2001). $N_2O$ emissions from nitrification are calculated as a function of soil $NH_4^+$ concentration, temperature, pH, texture and soil moisture. $N_2O$ from denitrification is calculated as a function of soil $NO_3^-$ concentration, soil moisture, texture and heterotrophic $CO_2$ respiration rate. $N_2O$ emissions from denitrification increases

exponentially when the WFPS exceeds the texture related threshold value and levels off as the soil approaches saturation. The model can simulate different types of tillage (i.e. plowing, tandem disk and field cultivator). Depending on the type of tillage, the decomposition of litter and SOM (active and slow) pools are increased by a specific factor for a period of one month, and a fraction of above-ground residues is transferred to surface litter and top soil layer. Tillage also impacts soil temperature and water dynamics indirectly because the model assumes that precipitation intercepted by surface litter and living biomass

evaporates before entering soil. The presence of surface litter insulates the soil from air temperature fluctuations.

If site level measurements of soil hydraulic properties required for DayCent are not available, they are calculated through the PTF from Saxton et al. (1986) and are static throughout the simulations. The PTF uses soil texture to calculate FC, WP, bulk density and Ksat. The soil water model simulates unsaturated water flow using Darcy's equation, runoff, snow dynamics, and the effect of soil freezing on saturated water flow (Pannkuk et al., 1998). DayCent has been shown to reliably model soil

water content, N mineralization and $N_2O$ emission rates from different soil types and management practices (Kelly et al., 2000; Parton et al., 2001). Del Grosso et al. (2002) provides an extensive overview of validate results for DayCent.

## 2.4 Experimental sites

Four experimental sites were selected in which the effects of tillage and no-tillage on $N_2O$ emissions were studied (Table 1 and Table 2). The sites were selected based on the availability of observational data and treatment combination of tillage and

no-tillage.

The first study site is located at the Agricultural Research Development and Education Center (ARDEC) near Fort Collins, CO (40° 39'6" N, 104° 59'57" W; 1555 m asl). It was initiated in 1999 on a clay loam soil (fine-loamy, mixed, mesic Aridic Haplustalfs), that was continuously cropped with maize (*Zea mays L.*) for six years. Shortly before sowing, fertilizers (67 kg N ha$^{-1}$) were applied. The fields were sprinkler irrigated during the growing season. In the tillage treatment, fields were

tilled shortly before sowing, and with harvest, followed by tandem disking and then moldboard plowing to a depth of 25 to 30 cm. $N_2O$ emissions were measured three times per week during the growing season (2002-2006) with closed chambers. Soil moisture was measured two to three times per month during the growing season from 2003 to 2006. Soil organic carbon (SOC) was measured once in October 2005. A detailed description of the experimental site can be found in Halvorson et al. (2006).





The second study site is located at the University of Nebraska-Lincoln Agricultural Research and development Center,
Ithaca, NE (41° 9'43.3"N, 96° 24'41.4" W; 349 m asl). The experiment was established in 2002 on a silt loam soil that was
previously cropped with rain fed maize, soybean (*Glycine max (L.) Merr.*), oat (*Avena sativa L.*) and alfalfa (*Medicago sativa*
*L.*). Maize was grown continuously on the field after 2000. During the experiment, N fertilizers were injected to a depth of
10-15 cm, once during the growing season at various rates and compositions (Table 1). The soil in tillage treatments was tilled
before sowing and at harvest to a depth of 15-20 cm. The field was irrigated with varying irrigation amounts. $N_2O$ emissions
were measured from April 2011 through May 2016 monthly during the growing season using closed chambers. Soil moisture
was measured at varying intervals from one to five times per month between 2011 and 2015. SOC was measured in May 2001,
November 2010, and November 2014 for different depths (0-0.15, 0.15-0.30, 0.30-0.60, 0.60-0.90, 0.90-1.20 and 1.20-1.50
m). More information regarding the experimental study site is provided by Jin et al. (2017).

The third study site is the W.K. Kellogg Biological Station Long-Term Ecological Research (KBS LTER) experiment
located in Southwest Michigan (42° 24' N, 85° 24' W, 288 m asl) on loam soils (Typic Hapludalfs). The experiment was
established in 1988 on an agricultural field that had been tilled for at least 100 years before the experiment. The crop rotation
before 1995 consisted of maize followed by soybean. In 1995, wheat (*Triticum aestivum L.*) was planted after soybean, which
resulted in a maize-soybean-wheat rotation. After the harvest of wheat, the fields stayed bare until the fields were cropped with
maize again. This sequence was followed during the time span analyzed here (1989-2010). Different quantities of N-fertilizers
were applied at sowing and/or during the growing season for maize, during the growing season for wheat, and soybean did
not receive fertilizers (Table 1). The tillage treatment was tilled each year with sowing, then during the growing season and at
harvest, to a depth of 20 cm. The fields were not irrigated during the experiment. $N_2O$ emissions were measured once or twice a
month from June 1991 to October 2016 using closed chambers. Soil moisture was measured once per month during the growing
season from 1989 until 2017. SOC was measured annually since 1989 at multiple sampling depths. More information regarding
the experimental study site is provided by Grandy et al. (2006) and on the KBS LTER website (http://lter.kbs.msu.edu, *accessed*
*November 2018*).

The last study site is located in Boigneville, France (48° 33'N, 2° 33'E, altitude unknown) on a silt loam soil (Haplic
Luvisoil) (FAO, 1998). The experiment started in 1970 that had been tilled to 30 cm depth annually. During the experiment, the
site was cropped with a maize-wheat rotation, with maize being sown in April, harvested in October and directly followed by
tillage (20 cm for tillage treatments) and sowing of wheat. After harvest of wheat in April, the soil was left bare and was tilled
(20 cm) in November, and left fallow until planting maize in the next growing season. This sequence was followed during the
time span analyzed here (2003-2004). During the experiment, the maize received N-fertilizers in May and wheat in February
and April (Table 1). The fields were irrigated between the end of June and July. $N_2O$ emissions were measured on average
every three weeks using closed chambers. Soil moisture was not measured. Soil organic carbon was measured twice in 2003
and once in 2004 at various depths. More information regarding the study site can be found in (Oorts et al., 2007).




**Table 1.** Overview of experimental sites selected for the study.

| Location | Years of Experiment | Soil texture | Land use | Observations Timespan (average freq. in growing season) | Reference |
|---|---|---|---|---|---|
| **Boigneville, France** | 1971–2004 | Silt loam | Maize-Wheat | $N_2O$: 2003-2004 (three weeks) | Oorts et al. (2007) |
| **Fort Collins, Colorado** | 1999–2006 | Clay loam | Maize | $N_2O$: 2003-2006 (three days) | Halvorson et al. (2006) |
| **Hickory Corners, Michigan** | 1989–2010 | Loam | Maize-Wheat-Soybean | $N_2O$: 1991-2016 (2 weeks) | Grandy et al. (2006) |
| **Mead, Nebraska** | 2001–2015 | Silt loam | Maize | $N_2O$: April 2011-May 2016 (2 weeks) | Jin et al. (2017) |




**Table 2.** Overview of observed input data and LPJmL input data

| Site | Observed data Fertilization Amount (g N m⁻²) | Day of year | Tillage Day of year | Growing Season Sowing Day of year | Harvest Day of year | Soil Pools Soil C g C ** | Soil N g N ** | LPJmL data Fertilization Amount (g N m⁻²) | Day of year | Tillage Day of year | Growing Season Sowing Day of year | Harvest Day of year | Soil Pools Soil C g C** | Soil N g N ** |
|---|---|---|---|---|---|---|---|---|---|---|---|---|---|---|
| France | 15.8 | 131 | 301 | 107 | 282 | 4553.3 | 450.3 | 10.4 | 122 | 122 | 122 | 297 | 3827.8 | 297.0 |
| | | | | | | | | 10.4 | 191 | 303 | | | | |
| Colorado | 6.7 | 118 | 30 | 118 | 288 | 6092.0 | 460.4 | 8 | 123 | 123 | 123 | 249 | 6267.7 | 335.7 |
| | | | 109 | | | | | 8 | 188 | 270 | | | | |
| | | | 119 | | | | | | | | | | | |
| | | | 330 | | | | | | | | | | | |
| Michigan | 3.3 | 135 | 136 | 128 | 293 | 9834.2 | 1148.2 | 7.155 | 125 | 125 | 125 | 238 | 6188.4 | 760.4 |
| | 12.3 | 179 | 139 | | | | | 7.155 | 175 | 251 | | | | |
| Nebraska | 20.2 | 165 | 114 | 123 | 270 | 1762.2 | 1529.14 | 8.4 | 124 | 124 | 124 | 234 | 6267.7 | 335.7 |
| | | | | | | | | 8.4 | 177 | 131 | | | | |

- The data are for the years where maize is grown, and vary between years.

** Size of pools are given for soil depth: France, Colorado, Michigan and Nebraska are from 0 to 0.2, 0.2, 1.0 and 1.5 respectively.





### 2.5 Management information

#### 2.5.1 LPJmL standard setup using global input data

In the LPJmL.G.Orig scenario, all management information as well as soil C and N-pools were used as within the default global simulation of LPJmL (Table 3). The amount of mineral and organic fertilizers was provided by the global gridded crop

model intercomparison (Elliott et al., 2015) of the Agricultural Model Intercomparison and Improvement Project (AgMIP, Rosenzweig et al., 2013). It is based on global, gridded data sets for each crop (Mueller et al., 2012; Potter et al., 2010). Fertilizer is assumed to consist of 50% $NO_3^-$ and 50% $NH_4^+$. If fertilizer input is low ($\leq$ 5.0 gN m$^{-2}$), all is applied at sowing. Otherwise, only half of the fertilizer is applied at sowing and the remainder is applied when the phenological stage fraction (unitless) of the crop reaches 0.4 (Von Bloh et al., 2018). Irrigation events occur when the fractional soil moisture of the water

holding capacity (unitless) is below an irrigation threshold value of 0.7 for maize (Jägermeyr et al., 2015).

In the experiments with tillage, tillage occurs twice a year; once at sowing and once at the day of harvest. Sowing dates are calculated internally following Waha et al. (2012). Thereby, the sowing dates are calculated based on a set of rules depending on crop specific thresholds and climate. Here, the sowing date depends on a crop-specific temperature threshold (i.e. 14 °C for maize; Waha et al., 2012).

The size of the C and N pools are calculated internally during the spinup (5000 years) of the natural vegetation and land-use history. The land-use history is simulated as with DayCent, in order to establish a comparable starting point when the simulations for the experiments are conducted. Thereby, the spin-up is followed by a simulation of historical land-use change to account for effects on the pools based on the best available information of land management.

#### 2.5.2 LPJmL detailed setup using observed input data

Site-specific observed information for all management inputs as well as soil C and N pools were prescribed for simulation LPJmL.D.Orig (Table 3). For practical reasons, irrigation water was added to precipitation to enable the specification of the amount and the timing of irrigation events. This mimics a sprinkler irrigation technique as part of the irrigation water is intercepted by the canopy. As the current implementation of soil layers and tillage in LPJmL does not allow for distinguishing more detailed tillage types other than conventional tillage and no tillage, we ignored tillage activities that were less intensive

(e.g. "shredding"). In order to specify the growing season, phenological heat unit requirements and base temperatures were parameterized so that the simulated harvest dates were matching the reported harvest dates.

The soil C and organic N pools from the simulations were scaled to the observed values. This was done twice, once at the introduction of land-use during spin-up and once at the start of the treatment of the experimental site. If observations were not available for the start of the experiment, the first available observation was taken, assuming that pool sizes remained stable

over that time period. The pools (P) at each site were scaled as in equation 1:

$$P_{(cor,l)} = P_{(sim,l)} * \frac{Total_{(obs)}}{Total_{(sim)}} \tag{1}$$





Where $P_{(cor)}$ are the scaled carbon or nitrogen content of the soil pools (g C or N m$^{-2}$) in layer $l$ of the experimental site and $P_{(sim)}$, the simulated amounts of C or N contained in the soil and litter pools of the different layers $l$ of the experimental site. Total$_{(obs)}$ and Total$_{(sim)}$, are the total of C or N contained in the soil and litter pools summed over the different layers ($l$)
for which observational data of soil organic C and N were available (in g C or g N m$^{-2}$, respectively) of the experimental site. The differences between simulated and observed input data are depicted in Table 3.

### 2.6   LPJmL experimental simulations

Agricultural management consists of several practices. To analyze the importance of individual management aspects, we conducted a set of simulations as in LPJmL.D.Orig but ignored one site-specific management practice and replace it with the
global assumption as in LPJmL.G.Orig (Table 3). As an example: LPJmL.D.Orig-F, refers to the simulation where all management information are as in the LPJmL.D.Orig, except for the fertilizer information. Instead, the amount, timing and type of fertilizers were used as in LPJmL.G.Orig. Other experimental simulations refer to: LPJmL.D.Orig-I, LPJmL.D.Orig-GS, LPJmL.D.Orig-PS and LPJmL.D.Orig-T, that use the management information as in LPJmL.D.Orig, except for irrigation (I; timing and amount), growing season (GS; sowing- and harvest days), C and N pool sizes (PS) and the timing of tillage (T)
respectively. The naming of the simulation consists of three parts: 1) model used (LPJmL), 2) the experiment conducted (e.g. I, GS or PS) and 3) whether it includes modifications ("Mod"; see 2.7) or not ("Orig").

### 2.7   Model modifications

Lutz et al. (2019a) found that LPJml overestimates N$_2$O emissions. Because of the importance of soil moisture for N$_2$O emissions, we tested if modifying the simulation of soil moisture can contribute to improving the simulation of N$_2$O emission.
We modified the model with respect to the treatment of the residue cover of the soil in no-tillage systems and with respect to changing the soil parameterization.

As the soil covered by residues under no-tillage practices in LPJmL simulations is very high and thus leads to high soil moisture levels throughout the year (as soil evaporation is reduced and infiltration is enhanced), we tested modifications of the relevant functions for this aspect. To this end, we tested modifications of the parameters that translate litter amounts
into soil cover (Gregory, 1982) and those that determine how long the soil is covered with residues. Rather than changing well-established functions on litter decomposition (Schlüter et al., 2018), we modified the parameter on bioturbation that was introduced by Lutz et al. (2019a) and tested its effects on the reduction of the residue cover of the soil.

Lutz et al. (2019a) used an average value of 0.006 (m$^2$ g$^{-1}$) (falsely described as 0.004 in their publication, but used so in the code: https://doi.org/10.5281/zenodo.2652136) to translate litter biomass into a fraction of soil being covered with
residues, which was applied to all litter, neglecting variations in surface litter for different materials. The bioturbation rate was increased from 0.19% day$^{-1}$ to 0.63% day$^{-1}$ to account for the surface litter being transferred to the incorporated litter pool per day (equivalent to an annual bioturbation rate of 90%, versus 50% as assumed previously).

High N$_2$O emissions can also result from biases in the parameterization of hydraulic properties. For example, small differences between FC and WSAT lead to frequent triggering of denitrification. To study the role of soil moisture for causing

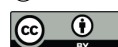

**Table 3.** Overview of management data used in LPJmL.D.Orig, LPJmL.G.Orig, Daycent and experimental runs.

| Management information | LPJmL.D. | LPJmL.G. | | Experimental runs LPJmL.D.Orig | | | | Daycent |
|---|---|---|---|---|---|---|---|---|
| | Orig | Orig | -F* | -I* | -GS* | -PS* | -T* | |
| Fertilizer (amount, type, timing) | Observed data | LPJmL Data | LPJmL data | Observed data | Observed data | Observed data | Observed data | Observed data |
| Irrigation (amount, timing) | Observed data | LPJmL data | Observed data | LPJmL data | Observed data | Observed data | Observed data | Observed data |
| Growing season | Observed data | LPJmL data | Observed data | Observed data | LPJmL data | Observed data | Observed data | Observed data |
| Tillage | Observed data | LPJmL data | Observed data | Observed data | Observed data | Observed data | LPJmL data | Observed data |
| Soil C and N pool | Observed data | LPJmL data | Observed data | Observed data | Observed data | LPJmL data | Observed data | Observed data |

* Experimental runs; all management information is as in the Detail setting, except for one scenario. For example, LPJmL.D-F* excludes fertilization information. The other settings exclude information on irrigation (LPJmL.D-I*), growing season (LPJmL.D-GS*), Pool N and C sizes (LPJmL.D-PS*), and tillage (LPJmL.D-T*).





deviations in tillage effects on $N_2O$ emissions, we analyzed if the parameterization of the hydraulic properties causes the overestimation in soil moisture. As observational data on the hydraulic properties are lacking, we here compared the hydraulic properties in relation to soil moisture from DayCent.

## 2.8 Analyses

### 2.8.1 $N_2O$ emissions

As $N_2O$ emissions are characterized by a high temporal variability, we analyzed two different aggregation levels: annual averages of $N_2O$ emissions and emissions of individual days within the year. We analyzed each tillage type ($tt$, i.e. conventional tillage and no-tillage) separately ($N_2O_{tt}$, equation 2) and differences between the two for both aggregation levels ($N_2O_{diff,year}$; equation 3 and $N_2O_{diff,day}$; equation 4).

$$N_2O_{tt} = \frac{\sum_{day=1}^{n} N_2O_{day,tt}}{n_{tt}} \quad (2)$$

$N_2O_{tt}$ is the annual average of simulated and observed $N_2O$ emissions (in g N ha$^{-1}$ d$^{-1}$) of $tt$ (tillage type: conventional tillage (*till*) or no-tillage (*notill*)), and $n_{tt}$ is the number of days with $N_2O$ emissions simulated or observed in the year of $tt$. Thereby, $n_{tt}$ equals all 365 days in the simulations and for the observations $n_{tt} < 365$ as observations are not available for every day in the year. We thus assumed that the scarcer observations still represent the full year's dynamics.

The differences in $N_2O$ emissions on annual average ($N_2O_{diff,year}$) were calculated as in equation 3:

$$N_2O_{diff,year} = \frac{\sum_{day=1}^{n} N_2O_{day,notill}}{n_{notill}} - \frac{\sum_{day=1}^{n} N_2O_{day,till}}{n_{till}} \quad (3)$$

Where $N_2O_{day,notill}$ and $N_2O_{day,till}$ are daily $N_2O$ emissions in g N ha$^{-1}$ d$^{-1}$ for all the days in the year and $n_{notill}$ and $n_{till}$ the number of days with $N_2O$ emissions simulated or observed in the year for no-tillage and tillage, respectively.

The differences in $N_2O$ emissions for individual days were calculated as in equation 4:

$$N_2O_{diff,day} = N_2O_{notill} - N_2O_{till} \quad (4)$$

Where $N_2O_{notill}$ and $N_2O_{notill}$ are daily emissions in all years.

The relative difference (RD in %) of no-tillage to conventional tillage was calculated as in equation 5:

$$RD = \left( \frac{\sum_{day=1}^{n} N_2O_{notill}}{\sum_{day=1}^{n} N_2O_{till}} \right) * 100(\%) \quad (5)$$

Where $N_2O_{notill}$ and $N_2O_{till}$ are daily $N_2O$ emissions in g N ha$^{-1}$ d$^{-1}$ for all the days in the year and $n$ is the number of days with $N_2O$ emissions simulated or observed.





### 2.8.2 Soil moisture

For the analyses of soil moisture, we focused on the first 0.2 m of the soil, which is the tillage-affected layer. We analyzed the experimental site in Nebraska as this site had the most observations of soil moisture compared to the other experimental sites. As $N_2O$ emissions are regulated by the WFPS in both LPJmL and DayCent, we normalized the soil moisture content and hydraulic properties to porosity ($W_{SAT}$ in mm). The WFPS (fraction) is calculated as in equation 6:

$$WFPS = \frac{W}{W_{SAT}} \tag{6}$$

where $W$ is the volumetric soil water content (mm). The $WFPC_{FC}$ (fraction) and $WFPC_{WP}$ (fraction) are the field capacity and wilting point values normalized to WFPS as in equations 7 and 8:

$$WFPC_{FC} = \frac{W_{FC}}{W_{SAT}} \tag{7}$$

$$WFPC_{WP} = \frac{W_{WP}}{W_{SAT}} \tag{8}$$

The $W_{FC}$ and $W_{WP}$ are the water content at field capacity and wilting point, respectively.

### 2.8.3 Evaluation metrics

To quantify the performance of simulated $N_2O$ emissions, we conducted an analyses of coincidence (equation 9) and an analysis of association (equation 10), following Smith and Smith (2007). Therefore, we calculated the deviation between simulated and observed values were by the root mean squared deviation (RMSD in g N ha$^{-1}$ d$^{-1}$) for the different sites as in equation 9:

$$RMSD = \sqrt{\frac{\sum_{i=1}^{n}(O_i - S_i)^2}{n}} \tag{9}$$

$O_i$ is the average observed $N_2O$ emission (in g N ha$^{-1}$ d$^{-1}$) of year $i$ and $S_i$ the average simulated value of $N_2O$ emission (in g N ha$^{-1}$ d$^{-1}$) of year $i$ and $n$ is the total number of valid value pairs for comparison.

To describe how well the dynamics in the observations were captured in the simulations, we calculated the degree of association ($r$) as in equation 10:

$$r = \frac{\sum_{i=1}^{n}(O_i - \overline{O})(S_i - \overline{S})}{\sqrt{\sum_{i=1}^{n}(O_i - \overline{O})^2 \sum_{i=1}^{n}(S_i - \overline{S})^2}} \tag{10}$$

Where $\overline{O}$ and $\overline{S}$ are the average observed and average simulated value respectively over all years (in g N ha$^{-1}$ d$^{-1}$). The significance of $r$ corresponds to the tests, null hypothesis: $r=0$.





The mean bias ($MB$ in fraction) was calculated as in equation 11:

$$MB = \frac{\overline{O}}{\overline{S}} \tag{11}$$

For soil moisture, the $RMSD$ and $r$ were calculated as well. However, there we focused on one site and calculated the average $RMSD$ and $r$ over all the years, as not much variation in soil moisture is expected between the years.

## 3  Results and discussion

### 3.1  Importance of management information

#### 3.1.1  Tillage effects on N$_2$O emissions

**Annual averages**

The N$_2$O emissions were overestimated in the LPJmL.G.Orig experiment when analyzing yearly averages of the different sites (Fig. 1 A). This effect was stronger for simulated emissions under no-tillage (RMSD=36.2 g N ha$^{-1}$ d$^{-1}$, r=-0.07) than under tillage (RMSD= 23.6 g N ha$^{-1}$ d$^{-1}$, r=-0.31). DayCent was closer to the observed values for both tillage (RMSD=7.60 g N ha$^{-1}$ d$^{-1}$, r=0.67) and no-tillage (RMSD=4.61 g N ha$^{-1}$ d$^{-1}$, r=0.66). For the full statistical analyses, we refer to Table A1 in
the Appendix.

Using detailed site-specific management information in LPJmL (LPJmL.D.Orig) improved the correlation between the observed and simulated values (Fig. 1 B). The simulated N$_2$O emissions under no-tillage deviated more from the observed values (RMSD= 38.9 g N ha$^{-1}$ d$^{-1}$, r=0.36), as the N$_2$O emissions were still overestimated. This held for the simulated N$_2$O emissions resulting under conventional tillage as well (RSMD=31.7 g N ha$^{-1}$ d$^{-1}$, r=0.34).

When analyzing the effect of tillage (difference between no-tillage and tillage), LPJmL.G.Orig showed an increase in emissions with no-tillage (Fig. 2 A), and LPJmL.D.Orig showed both an increase and decrease with no-tillage (Fig. 2 B). On average, no-tillage increased N$_2$O emissions by 59.5% in LPJmL.G.Orig, and 22.4% in LPJmL.D.Orig across all sites and years.In observations, no-tillage decreased N$_2$O emissions on average by 16.0% and DayCent shows a reduction of 24.3%. However, observations across the different sites showed, that no-tillage can have very different effects on N$_2$O emissions.
In Boigneville and Michigan, N$_2$O emissions increased under no-tillage (49.3% and 15.7% respectively), whereas it decreased in Colorado (by 9.01%) and Nebraska (by 29.2%). LPJmL.D.Orig reproduced the observed differences in tillage better (RSMD=12.0 g N ha$^{-1}$ d$^{-1}$, r=0.48) than LPJmL.G.Orig (RSMD=18.0 g N ha$^{-1}$ d$^{-1}$, r=-0.16), see also Fig. 2. Yet, both versions mainly projected an increase in N$_2$O emissions from no-tillage practices. DayCent results were closer to the observed values, but slightly underestimated the effects of no-tillage on N$_2$O emissions (RMSD= 4.96 g N ha$^{-1}$ d$^{-1}$, r=0.34).

**Daily emissions**

The simulations with different management information showed that these are relevant for the simulated tillage effects on N$_2$O emissions on individual days (Fig. 3). On average, more accurate information on management improved the simulations of differences between conventional and no-tillage systems in LPJmL except for the site in Colorado. However, there was



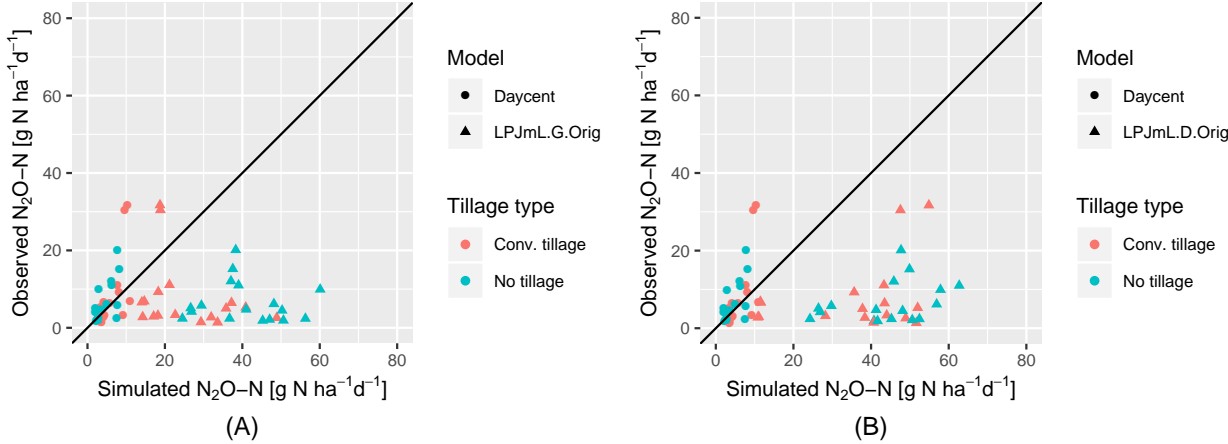

**Figure 1.** Comparison of observed and simulated yearly averages of $N_2O$ emissions by tillage type and models LPJmL.G.Orig (A), LPJmL.D.Orig (B) and DayCent. Data refer to all four sites and years of the experiments. Each point represents the average of all measured daily values within one year and tillage treatment. Tillage types are indicated by different colors.

no clear pattern between the different experimental runs of LPJmL (Fig. A1 in Appendix A). None of the simulations with
partial usage of detailed management information (Table 3) performed clearly better or worse between the LPJmL simulations. There were only small differences in the distribution of no-tillage effects on $N_2O$ emissions as well as between the averages. The observations showed that no-tillage both increased (Boigneville, Michigan) and decreased $N_2O$ emissions (Colorado, Nebraska) on average, as well as on the individual days. The negative effects were reproduced by DayCent in Colorado and Nebraska. The positive and negative effects were reproduced by LPJmL.D.Orig as well, except in Michigan. LPJmL.G.Orig
however, only reproduced the increase in $N_2O$ emissions in Michigan (Fig. 3).

In Colorado, observations showed a decrease in $N_2O$ emissions under no-tillage compared to conventional tillage. In contrast, LPJmL.D.Orig and LPJmL.G.Orig showed an increase in emissions with no-tillage, whereas the observed decrease was well captured by DayCent. In Boigneville, the increase in $N_2O$ emissions under no-tillage was well captured by LPJmL.D.Orig. DayCent and LPJmL.G.Orig did not capture the increase in $N_2O$ emissions with no-tillage. In Nebraska, LPJmL.D.Orig and
DayCent agreed with observations that no-tillage decreases $N_2O$ emission. In Michigan, no-tillage resulted mainly in an increase in emissions in LPJmL, which can also be found in the observations but not in DayCent simulations.

For all sites, LPJmL showed a high variability in $N_2O$ emissions between days (Fig. 3 and Table A1 in Appendix A). The interquartile ranges from LPJmL simulations were often much wider compared to observations and DayCent simulations. Hence, the variability of no-tillage effects on daily $N_2O$ emissions was overestimated. DayCent tended to underestimate the
variability of $N_2O$ emissions between days (Table A1 in Appendix A).

In LPJmL, the $N_2O$ emissions from no-tillage were entirely caused by changes in denitrification, whereas no-tillage mainly caused decreases on $N_2O$ emissions from nitrification (Fig. A2 in Appendix A). This can be explained by higher soil



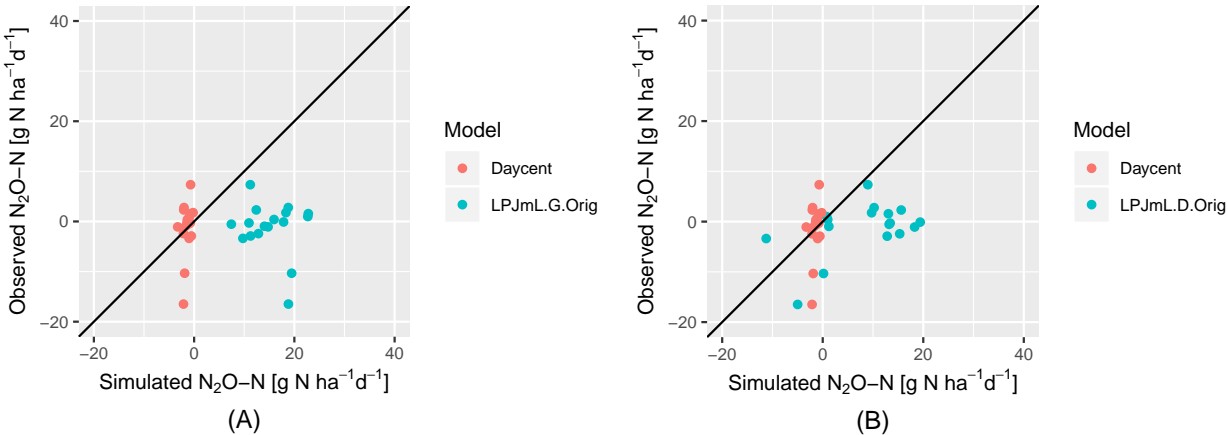

**Figure 2.** Comparison of observed and simulated effects after converting to no-tillage. The data refer to yearly averages of N$_2$O emissions and models LPJmL.G.Orig (A), LPJmL.D.Orig (B) and DayCent, of all four sites and years of the experiments.

moisture levels with no-tillage in LPJmL. In general, higher soil moisture levels trigger N$_2$O emissions from denitrification (anaerobic process), whereas nitrification is decreased (aerobic process). In DayCent, no-tillage mainly decreased N$_2$O emis-
sions emitted from nitrification and had little effects on denitrification.

### 3.2 Soil hydrology and model modifications

#### 3.2.1 Soil hydrology

The soil moisture (WFPS) simulated by LPJmL.D.Orig for no-tillage in Nebraska, is high compared to the observed values (RMSD= 0.24 (unitless), r= 0.28) (Fig. 4). DayCent was closer to the observed values for no-tillage (RMSD= 0.10 (unitless),
r=0.50) and tillage (RMSD= 0.11 (unitless), r=0.49). After modifying the parameters for surface litter and the hydraulic properties, the simulated soil moisture in the experiment LPJmL.D.Mod was closer to the observed values and simulation results from DayCent (Fig. 4). These combined effects showed the best performance for both tillage (RSMD=0.12 (unitless), r=0.33) and no-tillage (RSMD=0.14 (unitless), r=0.48), compared to implementing the modifications separately (Table 4). The dynamics in soil moisture simulated in the experiment LPJmL.D.Mod better reflected the dynamics simulated by DayCent.
For instance, after October, a decrease in soil moisture was simulated by DayCent (and measured) which was previously not captured by LPJmL.D.Orig. In LPJmL.D.Orig, soil moisture was mostly stationary around FC, which in LPJmL.D.Mod was only the case from April to June.

Although the simulation of soil moisture was improved with the modified settings, LPJmL simulations still overestimated soil moisture in comparison to observations.

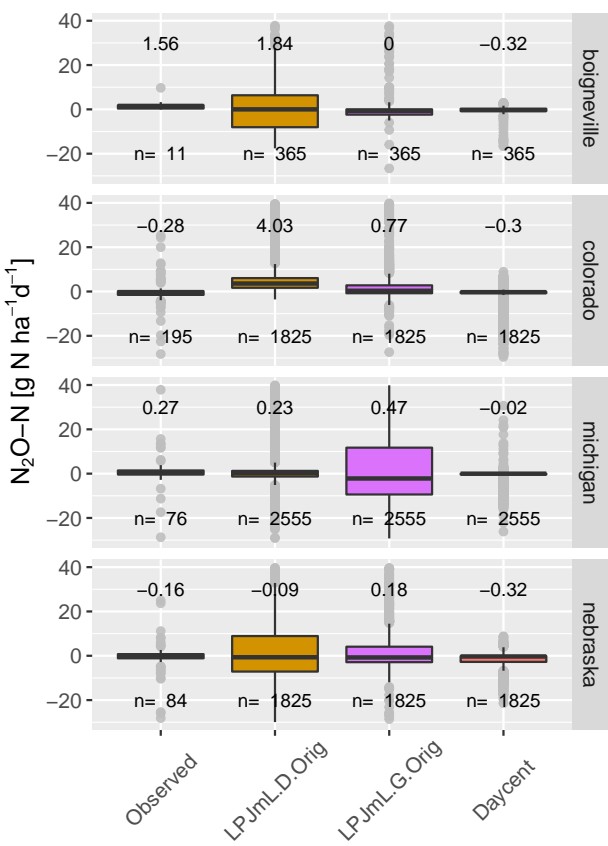

**Figure 3.** Effects of no-tillage on $N_2O$ emissions on individual days (and on average), including the original LPJmL settings, the observations and simulated values by DayCent.

**Table 4.** Performance of Daycent, and LPJmL compared to soil water observations in Nebraska. The results are shown for both conventional tillage and no-tillage

|  | $RMSE$ | | $r$ | |
| --- | --- | --- | --- | --- |
|  | **Conv. tillage** | **No tillage** | **Conv. tillage** | **No tillage** |
| **LPJmL.D.Orig** | 0.21 | 0.24 | 0.10 | 0.28 |
| **Bioturbation** | 0.20 | 0.22 | 0.19 | 0.40 |
| **Parameter residue cover** | 0.19 | 0.24 | 0.20 | 0.32 |
| **Hydraulic properties Daycent** | 0.15 | 0.18 | 0.07 | 0.23 |
| **LPJmL.D.Mod** | 0.12 | 0.14 | 0.33 | 0.48 |
| **Daycent** | 0.11 | 0.10 | 0.49 | 0.50 |



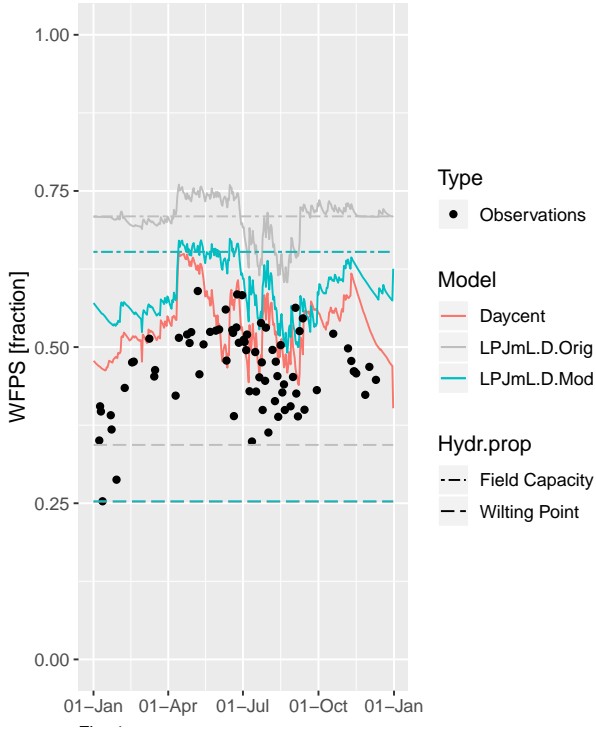

**Figure 4.** Observed and simulated soil moisture of no-tillage in the top soil (0-20 cm) in Nebraska.

### 3.2.2 Tillage effects on $N_2O$ emissions after modifications

**Yearly averages**

The modifications of the parameters for surface litter and the hydraulic properties improved the yearly tillage and no-tillage effects on $N_2O$ emissions across all the different sites (Fig. 5). The emissions under no-tillage (RSMD=18.1 g N ha$^{-1}$ d$^{-1}$, r=0.60) and under tillage (RSMD=16.3 g N ha$^{-1}$ d$^{-1}$, r=0.38) were much closer to the observed values than with the original hydrologic parameterization. Although the modifications improved the simulation of tillage and no-tillage, LPJmL.D.Mod still overestimated the changes in emissions when switching from conventional tillage to no-tillage systems (Fig. 5, Table A1). The modifications did not improve the simulation of $N_2O$ emissions after shifting to no-tillage (Fig. 6). Although the deviations of the absolute differences between tillage systems decreased, the correlation with observations was less well captured (RMSD=7.35 g N ha$^{-1}$ d$^{-1}$, r=-0.04), negating the improvements achieved through the consideration of detailed management information (LPJmL.G.Orig vs. LPJmL.D.Orig). The conversion to no-tillage systems increased $N_2O$ emissions by 13.0% in LPJmL.D.Mod. The increase in $N_2O$ emissions after shifting to no-tillage in the modified simulations was found across all sites in LPJmL.D.Mod, whereas DayCent showed decreases in $N_2O$ emissions across all sites at the yearly aggregation (Fig. 6). However, the observations showed both increases and decreases in $N_2O$ emissions after shifting to no-tillage for all sites at the yearly aggregation.





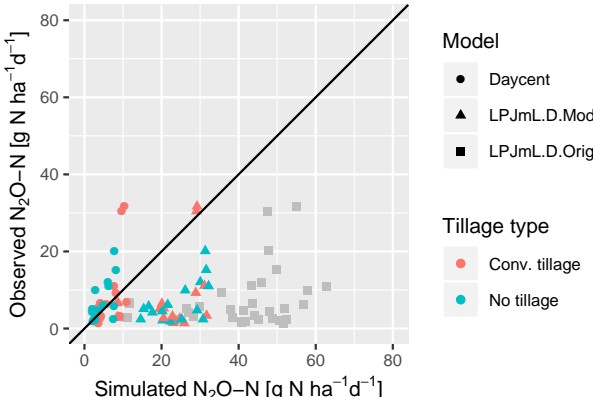

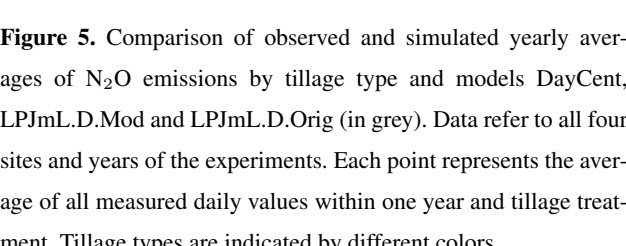

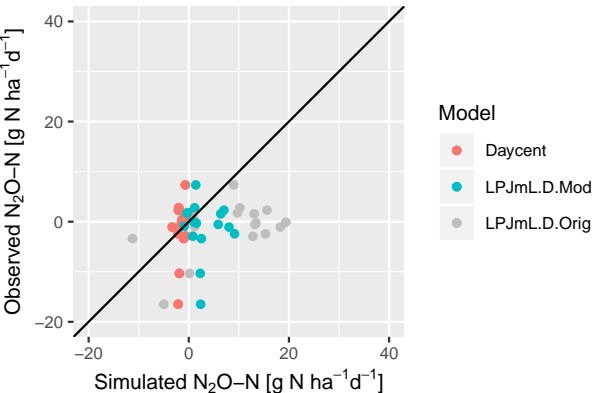

**Figure 5.** Comparison of observed and simulated yearly averages of $N_2O$ emissions by tillage type and models DayCent, LPJmL.D.Mod and LPJmL.D.Orig (in grey). Data refer to all four sites and years of the experiments. Each point represents the average of all measured daily values within one year and tillage treatment. Tillage types are indicated by different colors.

**Figure 6.** Comparison of observed and simulated effects after converting to no-tillage. The data refer to yearly averages of $N_2O$ emissions and models DayCent, LPJmL.D.Mod and LPJmL.D.Orig (in grey). Data refer to all four sites and years of the experiments.

**Daily emissions**

The modified hydrology (LPJmL.D.Mod and LPJmL.G.Mod), decreased the variability of no-tillage effects on $N_2O$ emissions of individual days in most LPJmL simulations (Fig. A3 in Appendix A). The interquartile ranges from daily $N_2O$ emissions simulated by LPJmL were more in agreement compared to the observations and DayCent, as the variability of no-tillage effects on $N_2O$ emissions is declined.

In the LPJmL.D.Mod experiment, simulated $N_2O$ emissions from no-tillage are now produced by both denitrification and nitrification (Fig. A2 in Appendix A). The increases in emissions from denitrification were smaller than in the LPJmL.D.Orig experiment and closer to the simulated values by DayCent in Boigneville and Nebraska. The emissions from nitrification increased by switching from conventional tillage to no-tillage systems, whereas they decreased in the LPJmL.D.Orig experiment. However, changes in nitrification remain small, compared to changes in denitrification.

**4  General discussion**

Detailed information on agricultural management improved the LPJmL simulation of $N_2O$ emissions produced by tillage and no-tillage, as well as of the effect of switching from conventional tillage to no-tillage systems. However, also with detailed information, LPJmL overestimated the $N_2O$ emissions. The overestimation is caused by too high simulated soil moisture, resulting in high fluxes from denitrification. After correcting for the overestimation in soil moisture, by modifying 1) the parameter that translate litter amounts into soil cover, 2) the parameter that determines the duration of the surface litter layer





and 3) hydraulic properties, the yearly averages of $N_2O$ emissions were closer to the observed values for tillage and no-tillage separately, but not for shifting from conventional tillage to no-tillage. However, the variability of no-tillage effects on $N_2O$ emissions between the days is now reduced in most of the LPJmL simulations and the interquartile ranges from LPJmL simulations are now in better agreement with observations and DayCent.

DayCent performed better in simulating tillage and no-tillage effects on $N_2O$ emissions on the yearly averages. However, DayCent tended to underestimate the overall effects and the inter annual variability of no-tillage on the emissions. DayCent mostly simulated a decrease in $N_2O$ emissions upon shifting to no-tillage. A major reason for this is that in DayCent conversion to no-tillage leads to increasing soil organic matter which is associated with decreased availability of mineral N. However, observations showed that no-tillage can also increase $N_2O$ emissions. For example, no-tillage can result in increased soil

moisture content which can promote $N_2O$ emissions from denitrification. DayCent simulations showed basically no response in $N_2O$ emissions from denitrification. On the other hand, conventional tillage can increase the decomposition rate of (soil) organic matter, through improved aeration of the soil. Increased decomposition leads to an increase of available N that can be transformed to $N_2O$ through nitrification and denitrification. The higher $N_2O$ emissions with conventional tillage in DayCent, indicates that the increase in decomposition rate of (soil) organic matter due to tillage, is dominant in comparison to the effect

of increased soil moisture-driven denitrification rate.

The overall better performance by DayCent likely reflects the years of model development and testing at this scale and previous application at these sites (except the site in Boigneville) (Campbell et al., 2014; Del Grosso et al., 2008; Yang et al., 2017), which enabled more accurate reproduction of observed $N_2O$ emissions. The testing of the model performance as well as improvements to reproduce observed $N_2O$ emissions has been conducted in several studies (Necpálová et al., 2015; Fitton

et al., 2014; Del Grosso et al., 2010). For example, model calibration has been conducted to test the model performance based on contributing parameters and key processes that affect $N_2O$ emissions. For instance, the maximum amount of $N_2O$ emissions produced during nitrification as well as the proportion of nitrified N that is lost as $N_2O$ can be specified. LPJmL, is developed for global-scale applications and is therefore usually not calibrated, as suitable calibration targets are typically not available at that scale.

The application of LPJmL at the experimental sites provided much insight into the deviations of the tillage effects on $N_2O$ emissions from observations. It enabled to use site-specific information on agricultural management, whereas missing information at global scale has to be supplemented with assumptions. As detailed information improved the simulation of tillage effects on $N_2O$ emissions, advancing the current state of information on agricultural management at the global scale could improve global estimates of tillage effects on $N_2O$ emissions. The study also highlighted the potential of improving the

simulation of $N_2O$ emission by improving soil moisture dynamics. Any modification to improve LPJmL5.0-tillage needs to be evaluated at the global scale, as LPJmL is typically applied at that scale (e.g. Heinke et al., 2019; Rolinski et al., 2018; Schaphoff et al., 2018). A first recommendation is to revisit the PTF used in LPJmL5.0-tillage. We saw in this exercise that LPJmL overestimated soil moisture independent of the tillage system. Although the modifications in residue cover improved the results on soil moisture, the most important modification was in the hydraulic properties resulting from the PTF. The modifications

still resulted in relatively high soil moisture contents, and therefore possibly still overestimations in $N_2O$ emissions. A reason





for this could be the relatively inefficient percolation of soil moisture to lower soil layers as soon as soil moisture is higher than FC.

N$_2$O emissions from denitrification increase exponentially when the WFPS exceeds a certain threshold value in LPJmL. This threshold value (which is around 0.8 of WFPS) is a proxy for assuming anaerobic conditions, and is static for all soil
texture types. However, finer-textured soils have lower gas diffusivity at a given WFPS than coarser textured soils (e.g. Del Grosso et al., 2000). In soils with lower gas diffusivity, denitrification is assumed to occur at lower levels of WFPS, because atmospheric O$_2$ may not diffuse into the soil fast enough to fully satisfy microbial demand (Parton et al., 1996). Threshold values for anoxic conditions that are soil texture type specific are currently not accounted for in LPJmL. In DayCent, the effect of gas diffusivity of different soil texture types is taken into account. An index of gas diffusivity is calculated based on
the WFPS, bulk density and FC, which is a proxy for pore size distribution and air filled pore space. This index influences the denitrification rate (i.e. lower diffusivity increases denitrification), N$_2$ to N$_2$O and NO$_x$ to N$_2$O ratios. Including such processes in LPJmL might improve simulated N$_2$O emissions. However, this would require suitable reference data in order to parameterize these processes well.

## 5  Conclusions

Previous findings have shown deviations between simulations with the LPJmL5.0-tillage model and the results from meta-analyses on global estimates of tillage effects on N$_2$O emissions. In this study, we tested LPJmL5.0-tillage at different experimental sites to study whether deviations in N$_2$O emissions result from a lack of detailed information on agricultural management and/or the representation of soil water dynamics. The results were compared to observed values of the experimental sites as well as to results of the field-scale model DayCent.

Adding site-specific information on agricultural management improved the simulation of N$_2$O emissions under conventional tillage and no-tillage practices, as well as changes in emissions from shifting from conventional tillage to no-tillage in LPJmL5.0-tillage. Although adding information on agricultural management improved the performance of LPJmL5.0-tillage, simulated N$_2$O emissions remained too high, due to a general bias in over-estimations of soil moisture. By modifying the parameters related to residue cover and the hydraulic properties as used in DayCent, the simulation of soil moisture and N$_2$O
emissions by LPJmL5.0-tillage improved substantially.

Generally, there is substantial uncertainty in simulating the effects of different tillage systems on N$_2$O emissions. DayCent performed better in simulating N$_2$O emissions under conventional tillage and no-tillage, but generally showed little response in N$_2$O emissions on changes in tillage practices. LPJmL5.0-tillage simulations reproduced a broader range of tillage effects on N$_2$O emissions, but tended to overestimate N$_2$O emissions in general. Modifications to the detail of management information
considered and soil hydrology could always only improve in one deficiency (bias or dynamics) but not in both.

This study confirmed that the deviations in N$_2$O emissions can be explained by both lacking detailed information on management and relative high soil moisture levels simulated by LPJmL5.0-tillage. Advancing the current state of information on agricultural management can thus improve global estimates of tillage effects on N$_2$O emissions. Furthermore, the repre-





sentation of soil water dynamics and $N_2O$ dynamics highlights the potential to improve LPJmL5.0-tillage. However, given the

limited skill to reproduce observed patterns in simulations with LPJmL5.0-tillage, the model currently does not lend itself to evaluating the impacts of different tillage systems on $N_2O$ emissions but requires further research on better representation of soil hydrology and its effects on $N_2O$ emissions.

*Code and data availability.* The LPJmL source code is publicly available under the GNU AGPL version 3 license. An exact version of the code described here and the R script used for post processing the data from the simulations conducted are archived under

https://doi.org/10.5281/zenodo.3592381 (Lutz et al., 2019b).



# Appendix A

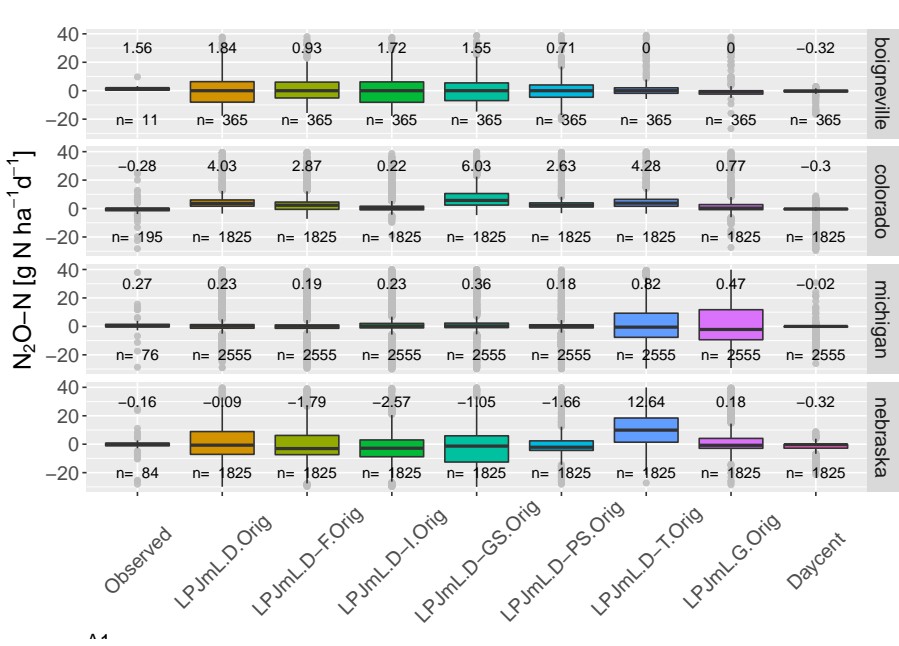

**Figure A1.** Effects of no-tillage on N$_2$O emissions on individual days by the different experimental simulations, including the original runs of LPJmL, the observations and simulated values by DayCent.



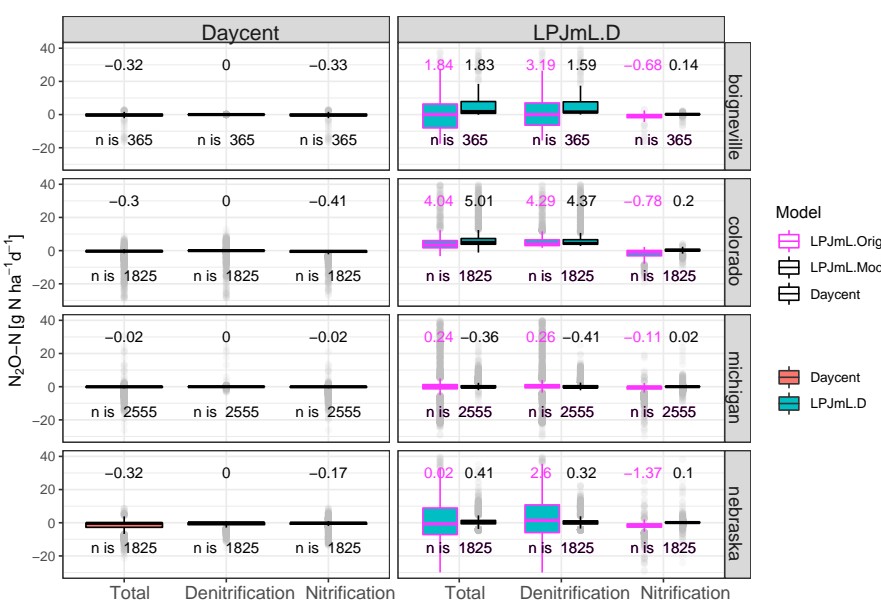

**Figure A2.** The relative share of N$_2$O emissions from nitrification and denitrification on individual days with no-tillage. The simulated values include the original (purple lines) and the modified (black lines) LPJmL settings. The simulated values by DayCent are also shown. Observed values are not available.

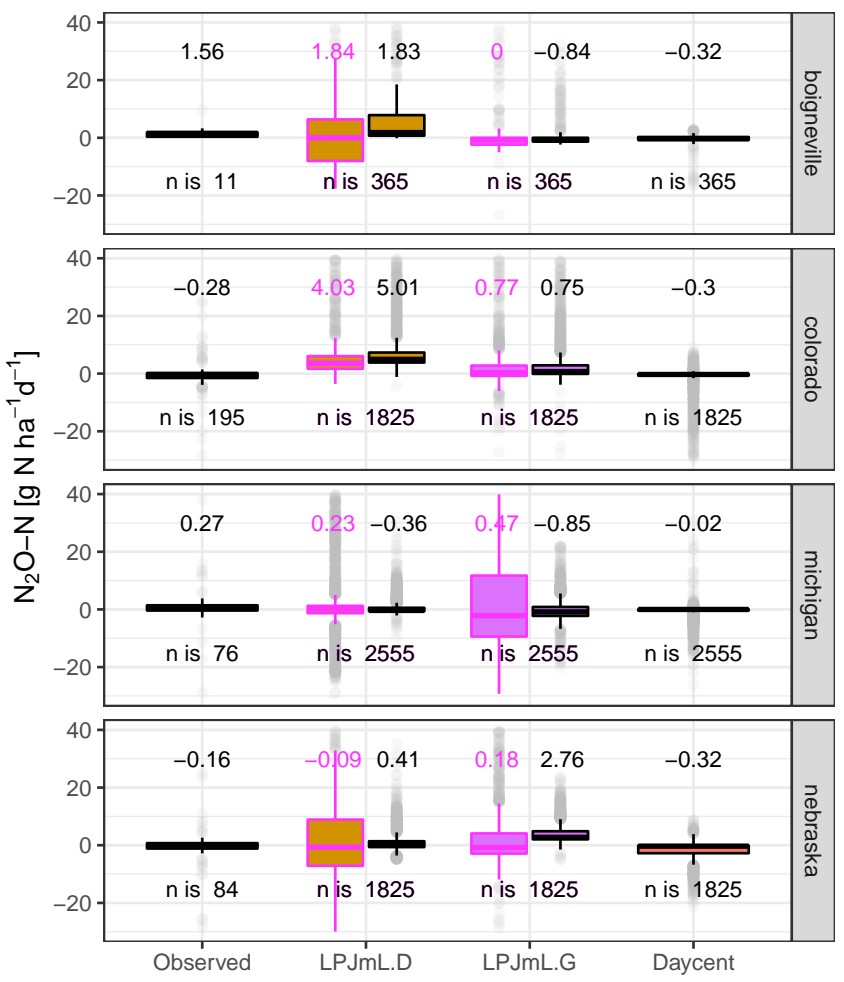

**Figure A3.** Effects of no-tillage on $N_2O$ emissions on individual days by the different experimental simulations, including the original- (purple lines) and the modified (black lines) simulations from LPJmL, the observations and simulated values by DayCent.





**Table A1.** Performance Daycent and LPJmL over all sites and years

| Models | Tillage type | RMSD | r | Sign. ($H_0: r = 0$) | Mean Bias | Standard Deviation |
|---|---|---|---|---|---|---|
| Daycent | Conv. tillage | 7.60 | 0.67 | 0.00 | 1.35 | 3.24 |
| Daycent | No tillage | 4.61 | 0.66 | 0.00 | 1.50 | 2.29 |
| LPJmL.G.Orig | Conv. tillage | 23.60 | -0.31 | 0.23 | 0.31 | 0.86 |
| LPJmL.G.Orig | No tillage | 36.20 | -0.07 | 0.80 | 0.16 | 0.51 |
| LPJmL.D.Orig | Conv. tillage | 31.70 | 0.34 | 0.18 | 0.22 | 0.59 |
| LPJmL.D.Orig | No tillage | 38.90 | 0.36 | 0.16 | 0.15 | 0.46 |
| LPJmL.G.Mod | Conv. tillage | 14.25 | -0.47 | 0.06 | 0.46 | 2.68 |
| LPJmL.G.Mod | No tillage | 13.86 | -0.07 | 0.79 | 0.34 | 2.41 |
| LPJmL.D.Mod | Conv. tillage | 16.30 | 0.38 | 0.13 | 0.37 | 1.11 |
| LPJmL.D.Mod | No tillage | 18.10 | 0.60 | 0.01 | 0.27 | 0.88 |
| Daycent | No tillage- Conv.tillage | 4.96 | 0.34 | 0.02 | 0.88 | 6.85 |
| LPJmL.G.Orig | No tillage- Conv.tillage | 18.00 | -0.16 | 0.55 | -0.08 | 1.17 |
| LPJmL.D.Orig | No tillage- Conv.tillage | 12.00 | 0.48 | 0.05 | -0.16 | 0.61 |
| LPJmL.G.Mod | No tillage- Conv.tillage | 7.17 | -0.33 | 0.19 | -0.60 | 2.21 |
| LPJmL.D.Mod | No tillage- Conv.tillage | 7.35 | -0.04 | 0.89 | -0.45 | 1.64 |





*Author contributions.* FL and CM designed the study in discussion with SDG and SO. FL conducted all model simulations and wrote the paper with support from CM. FL prepared all figures with support from SM. FL conducted the analyses with inputs from CM and JH. All authors edited the paper text.

*Competing interests.* The authors declare that they have no conflict of interest.

*Acknowledgements.* F.L., S.M and S.R. gratefully acknowledge the German Ministry for Education and Research (BMBF) for funding this work, which is part of the MACMIT project (01LN1317A). F.L also acknowledges the Huub and Julienne Spiertz Fund that enabled to visit the Colorado State University and USDA to collaborate on this research. Support for this research was also provided by the NSF Long-term Ecological Research Program (DEB 1832042) at the Kellogg Biological Station and by Michigan State University AgBioResearch.





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
