# Peer review of "The importance of management information and soil moisture representation for simulating tillage effects on $N_2O$ emissions in LPJmL5.0-tillage"

_Geoscientific Model Development, 2019_

## Referee Comment (RC1) · Anonymous Referee #1 · 30 Mar 2020

Comment on Lutz et al. - The importance of management information and soil moisture representation for simulating tillage effects on N2O emissions in LPJmL5.0-tillage

General comment

In this manuscript, Lutz et al. validate N2O emissions from the recent tillage version of the LPJmL model using field data from four sites and DayCent model outputs. They analyze the effect of management information and soil moisture representation on the model performance. Estimating the effects of tillage /no-tillage on GHG emissions the

regional and global scale is a topic of great scientific relevance. To establish a fair representation of that effect in biogeochemical models is important for upscaling and thus relevant, confirming the scientific interest of the paper.

The manuscript is well structured and used sound modeling experiments to compare the main effect of tillage in LPJmL to field sites, but also to investigate underlying causes of mismatches. The authors analyze the effect of exact management information on agricultural activities affects the simulation of tillage effects on N2O emissions LPJmL. They further show that the soil moisture content in LPJmL was overestimated at one site, and modifications of the hydraulic properties in the model improved soil water simulations and associated N2O emissions for this site. The manuscript shows that there is still room for improvement in process understanding related to tillage effects and its implementation in biogeochemical modeling.

A main weak point in the manuscript is that measured N2O fluxes were only in low temporal resolution. It is known that due to the sporadic nature of N2O fluxes and enormous flux variability, measurements in low temporal resolution lead to extremely high uncertainties (Barton et al., 2015). Measurements on a biweekly basis might miss entire N2O peaks which dominate annual fluxes.

Further several parameters in LPJmL (hydraulic properties) were not directly compared to measured data but with DayCent values (which can have their own issues). Still, the discontinuously measured values which are closer to DayCent values seem to justify this approach and also the results seem to prove this approach plausible.

Soil moisture was only investigated at one specific site, which reflects a certain pattern of properties, while it is not known if these results would hold for all sites.

There is one methodological issue in the modeling: Under completely anaerobic conditions, no more N2O will be emitted, instead there occurs complete reduction to N2. An exponential increase of N2O with soil moisture at very high moisture levels does not make sense.

I have a major concern about the input data on soil C and N stocks given in Table 3, which cannot be correct. In case the observational data was used exactly as depicted in the table, the concerned model runs need to be redone with corrected data.

Acknowledging the scientific value and the overall quality in the structure and the scientific quality of the paper, I would recommend to publish the manuscript in GMD after major revisions.

Barton, L., Wolf, B., Rowlings, D., Scheer, C., Kiese, R., Grace, P., Stefanova, K. and Butterbach-Bahl, K.: Sampling frequency affects estimates of annual nitrous oxide fluxes, Sci. Rep., 5(1), doi:10.1038/srep15912, 2015.

Specific comments

L 20: The sentence sounds a bit strange since the formation of GHGs is not a biogeochemichal process.

The introduction is well-written and gives a good overview on the topic.

L99: In this paragraph you describe one part of the curve, I would urge to describe the whole dependency of denitrification N2O on WFPS (full range of WFPS) as well as for N2O production during nitrification, since these relationships are very important for your research question.

Describe briefly the effects of incorporating residues at tillage on the soil pools.

L101: The N2O emissions from denitrification increases exponentially when the WFPS reaches a threshold value of $\geq$ 90%, as denitrification occurs only in oxygen deficit conditions (see also Krysanova and Wechsung, 2000).

Does not make sense to me. Do you mean decreases instead of increases? Or do you refer to increasing N2?

How does the relationship N2O produced by denitrification vs WFPS differ in your LPJmL version from DayCent?

[Figure]

L129: N2O emissions from denitrification increases exponentially when the WFPS exceeds the texture related threshold value and levels off as the soil approaches saturation. This suggests that there would be no decrease in N2O production at extremely high WFPS in Daycent? Which is missleading – please describe the whole relationship, also the decrease at high WFPS; (see Parton et al 2001)

Table 1: The order of the paragraphs describing the sites should be consistent with the site order in the tables.

Table 2: Soil pools: Units are incomplete: g C per what, g N per what?

When I look at the C-N ratios in Table 2, it becomes obvious that something must be wrong here. In Michigan (obs) this is rather low (8.6), however in Nebraska a Soil C/N ratio of 1.2 – this is impossible. Please check your values for C and N pools thoroughly. If really the values as written were used in the modeling, the respective runs must be redone with corrected values.

L 192: all is applied at sowing - So does this mean exactly at sowing date? (-> farmers would often do it rather 1-2 weeks later.)

L 195: 0.7 for maize, what about the other crops?

How do you explain the large differences in soil N between observed and simulated values (Table 2) in Michigan and Nebraska? The given depths are consistent between obs and sim, right?

Overall, the method section is clearly written and comprehensive enough to allow the reader to thoroughly understand the modeling experiment.

L276: "We analyzed the experimental site in Nebraska". Here I was quite surprised that the analyses of soil moisture was only performed at one site. I would add quite early, when you first mention the soil moisture analyses in the Intro or beginning of methods the specification "at one selected site".

L 298: "The significance of r corresponds to the tests, null hypothesis: r=0." The sentence as it is now, makes no sense since it is unclear to which test it refers. Please clarify.

L 315: In this paragraph, it would help the reader if you first briefly point out what tillage effect the observed data show, and only after that present the model results; then the reader has a chance to compare without again looking to the plot. As it is now, it is a bit cumbersome to read.

I would strongly urge to order: Observed results, then model results, then details.

L 336 Also in this paragraph sentences, I think for each it makes it easier to put first observed results (which I think shout be reference), then model results.

Fig 2: Is this the difference between no-tillage and tillage, N2Onotill – N2Otill? You could add this to the caption.

L 316: LPJmL.G.Orig showed an increase in emissions with no-tillage (Fig. 2 A)

L 326: use "more detailed" instead of different

L353: You give values for LPJmL only on no-tillage, but for DayCent tillage and no-tillage?

Fig 4: You use RMSD throughout the manuscript, but switch to RMSE here: Probably this is a typo so you meant RMSD and referred to WFPS as a fraction?

Fig 4: There is no need to extend the y axes from 0 to 1, which results in half of the area without information - instead the plot could be a bit larger.

How much differ the PTFs used in Daycent and LPJml?

L 433: Under completely anaerobic conditions, no more N2O will be emitted, instead there occurs complete reduction to N2. An exponential increase with soil moistures at very high moisture levels does not make sense.

Fig A1: Adding r values for each run would help to get insights about the degree of association. The mean and distribution give only a very limited picture.

Technical correctionsf

Methods: use past tense consistently e.g. L67 (whether the deviations are were)

L68:Four experimental sites with detailed information on management available were identified.

Table 2 I think that the numbers after the dot give the impression that these values would have this high precision. I would advise to round consistently.

L276: "we focused on the uppermost 0.2 m of the soil"

L288: "Therefore, we calculated the deviation between simulated and observed values were by the root mean squared deviation (RMSD in g N ha−1 d−1) for the different sites as in equation 9:"

The sentence makes no sense and needs to be revised.

L 318: "years.In observations" : Missing space after the point

Fig 3: Use capital letters for the sites, use n = 123 instead of n= 123; What is the number on top of each box? (-> add explanation to the caption); round consistently.

How would the relationship of WFPS vs N2O for nitrification and denitrification look like? Could you plot these for DayCent and LPJmL?

Result section: Use past tense consistently.
* * *

---

## Referee Comment (RC2) · Anonymous Referee #2 · 31 Mar 2020

Lutz and co-authors validated a model that estimates soil N2O emissions in tillage and not tillage agriculture against field experiments. They report that (1) the model perfornance is improved by using including site-specific land use information as a model input instead of global model estimates and that (2) the model perfomance bias (overestimation of emissions) is reduced by a better parametrisation of hydrological processes (to avoid an overestimation of soil moisture).

This is a well structured manuscript that makes important contributions to the incremental improvement of the LPJmL5.0-tillage model. The manuscript is well structured

and easy to read. Overall, I find the author work convincing and have only minor comments:

- I recommend removing the grey background and grid form the plots to improve the figures readablity. - General discussion and conclusion sections are almost of the same length and largely redundant.

---

## Author Comment (AC1) · 15 Jun 2020

**Dear Editor and Referees,**
Thank you for the thorough evaluation of our manuscript and the very helpful and detailed feedback. This is much appreciated. We were able to address all of the reviewers' points, which helped to improve the scientific rigor and presentation of our work in this paper.

In this letter we list the referees' comments, each point followed by our responses, and

the changes in the manuscript.

The responses and subsequent modifications to the manuscript have been derived in consultation with all co-authors.

Best regards,
Femke Lutz

**Referee # RC1**
**General comment:** In this manuscript, Lutz et al. validate N2O emissions from the recent tillage version of the LPJmL model using field data from four sites and DayCent model outputs. They analyze the effect of management information and soil moisture representation on the model performance. Estimating the effects of tillage /no-tillage on GHG emissions the regional and global scale is a topic of great scientific relevance. To establish a fair representation of that effect in biogeochemical models is important for upscaling and thus relevant, confirming the scientific interest of the paper.
The manuscript is well structured and used sound modeling experiments to compare the main effect of tillage in LPJmL to field sites, but also to investigate underlying causes of mismatches. The authors analyze the effect of exact management information on agricultural activities affects the simulation of tillage effects on N2O emissions LPJmL. They further show that the soil moisture content in LPJmL was overestimated at one site, and modifications of the hydraulic properties in the model improved soil water simulations and associated N2O emissions for this site. The manuscript shows that there is still room for improvement in process understanding related to tillage effects and its implementation in biogeochemical modeling.
A main weak point in the manuscript is that measured N2O fluxes were only in low temporal resolution. It is known that due to the sporadic nature of N2O fluxes and enormous flux variability, measurements in low temporal resolution lead to extremely high uncertainties (Barton et al., 2015). Measurements on a biweekly basis might miss

entire N2O peaks which dominate annual fluxes.

Further several parameters in LPJmL (hydraulic properties) were not directly compared to measured data but with DayCent values (which can have their own issues). Still, the discontinuously measured values which are closer to DayCent values seem to justify this approach and also the results seem to prove this approach plausible.

Soil moisture was only investigated at one specific site, which reflects a certain pattern of properties, while it is not known if these results would hold for all sites.

There is one methodological issue in the modeling: Under completely anaerobic conditions, no more N2O will be emitted, instead there occurs complete reduction to N2. An exponential increase of N2O with soil moisture at very high moisture levels does not make sense.

I have a major concern about the input data on soil C and N stocks given in Table 3, which cannot be correct. In case the observational data was used exactly as depicted in the table, the concerned model runs need to be redone with corrected data.

Acknowledging the scientific value and the overall quality in the structure and the scientific quality of the paper, I would recommend to publish the manuscript in GMD after major revisions.

*Thank you for the positive general assessment. We have revised the manuscript following your suggestions and by clarifying the methods. One concern that was noted in the general comment but not in the comments below was that measured N2O fluxes were only in low temporal resolution although N2O fluxes are known to have a high temporal variability.*

*The N2O emissions were observed 2-3 times/week during the growing season at the experimental site in Colorado and once or twice per week in Nebraska in April and May (there was a mistake in the manuscript, corrected now), which is a relatively frequent sampling regime (Barton et al., 2015). For the experimental site in Michigan and Boigneville the measurements were taken every two and three weeks respectively. For our study, we made use of data from existing experi-*

*mental sites that are mostly publicly available. These sites were chosen based on the availability of observational data and treatment combination of tillage and no-tillage (see also Line 156). We had no influence on the experimental design at those sites, including the amount of measurements that were taken. We point to the uncertainty related to sampling intensity in the discussion. We also more completely describe how soil water content influences N2O emissions, corrected the Table, and added in the introduction that only one site was used for soil water comparison. The other issues raised in these comments are addressed in responses below.*

**Referee comment 1:** L 20: The sentence sounds a bit strange since the formation of GHGs is not a biogeochemichal process. The introduction is well-written and gives a good overview on the topic.

*Answer 1: Thank you. The formation of GHGs is driven by the metabolism of microbes and can thus be considered a biochemical process. We revised the sentence to a more general statement: "The resulting changes in physical and chemical properties of the soil affect the living conditions of soil microbes and thus influence the formation of greenhouse gases (GHG)."*

*Changes in manuscript 1: The revised sentence can be found in Line 19-20.*

**Referee comment 2:** L99: In this paragraph you describe one part of the curve, I would urge to describe the whole dependency of denitrification N2O on WFPS (full range of WFPS) as well as for N2O production during nitrification, since these relationships are very important for your research question.

*Answer 2: We followed your suggestion and added this information to both the LPJmL and DayCent model:*

*LPJmL:* "*Nitrification is modeled after Parton et al (2001) with N2O emissions from nitrification being proportional to the nitrification rate. The nitrification rate depends on the water-filled pore space (WFPS), soil temperature, NH4+ and pH. Nitrification increases with higher levels of WFPS until it reaches the optimal WFPS value for nitrification (around 60%). Denitrification rates depend on the soil temperature, the availability of organic carbon and NO3-, and increases exponentially above 80% WFPS. As denitrification is an anoxic process, denitrification rates are negligible for levels of WFPS that are less than ~80%. Following the approach from Bessou et al. (2010), N2O emissions from denitrification are assumed to be proportional to the denitrification rate (11%).*"

*DayCent:* "*The N2O emissions from nitrification are proportional to the nitrification rate. Nitrification increases with water content, approaches maximum rates at WFPS of 50-60%, and declines after field capacity is exceeded (Hartman et al., 2018). The model also assumes that the portion of nitrified N that is lost as N2O increases with water content between wilting point and field capacity.*

*Denitrification rates increase exponentially when the WFPS exceeds the texture related threshold value (55-80%) and becomes static as the soil approaches saturation (around 90%) (Del Grosso et al., 2000). In addition to denitrification rates, N2O emissions also depend on the portion of N2 lost compared to N2O with the ratio of N2:N2O emissions assumed to increase as soils become wetter.*"

***Changes in manuscript 2:*** *The added information can be found in Line 104ff for LPJmL and Line 136ff for DayCent.*

**Referee comment 3**: Describe briefly the effects of incorporating residues at tillage on the soil pools.

***Answer 3:*** *Organic C and N in surface residues are incorporated into the soil through tillage and bioturbation (mixing of soil with residues by animal activity), forming*

*the incorporated litter pool. All pools are subject to decomposition, with the rates depending on temperature and moisture conditions. By incorporating residues into the soil column, decomposition is no longer a function of air temperature and the moisture of the above-ground litter, but of the temperature and moisture regime of the first soil layer (0-200mm). A fixed fraction of the decomposed litter is mineralized and emitted as CO2 whereas the remaining C is transferred to the soil C pools. The mineralized N is added to the NH4 pool. The soil C and N pools can thus be supplied with organic C and N contained in the surface litter through its incorporation into the soil resulting from tillage, followed by decomposition of soil C and mineralization of soil N. We briefly extended the description of the effects of tillage on litter pools and hence, the soil C and soil N pools.*

***Changes in manuscript 3:** The extended description of tillage effects on litter pools and hence, the soil C and soil N pools can be found in Line 97ff.*

**Referee comment 4:** L101: The N2O emissions from denitrification increases exponentially when the WFPS reaches a threshold value of 90% , as denitrification occurs only in oxygen deficit conditions (see also Krysanova and Wechsung, 2000). Does not make sense to me. Do you mean decreases instead of increases? Or do you refer to increasing N2?

***Answer 4:** Sorry for the confusion. We meant to say: Denitrification rates depend on the soil temperature, the availability of organic carbon and NO3⁻, and increases exponentially above 80% WFPS. As denitrification is an anoxic process, denitrification rates are negligible for levels of WFPS that are less than ∼80%. Following an approach by Bessou et al. (2010), N2O emissions from denitrification are proportional to the denitrification rates (von Bloh et al. 2018).*

***Changes in manuscript 4:** We added this information to Line 107-110.*

**Referee comment 5**: How does the relationship N2O produced by denitrification vs WFPS differ in your LPJmL version from DayCent?

*Answer 5: In LPJmL, the N2O produced with denitrification has very low values for WFPS of less than ~80% and increases exponentially until the soils reach saturation (see von Bloh et al. 2018). In DayCent, denitrification occurs in the interval 55%< WFPS<90% and increases exponentially but becomes static when soils approach saturation. In contrast to LPJmL, which assumes that portions of N2 and N2O lost from denitrification are constant, DayCent calculates the proportion of denitrification N gas losses that are in the form of N2 and N2O, depending on the oxygen availability (see also Parton et al., 2001 and Line 458ff in the manuscript).*

*Changes in manuscript 5: This information has been added in the model description within materials and method section, Line 104ff for LPJmL and Line 136ff for DayCent (see also our reply to commend 2).*

**Referee comment 6:** L129: N2O emissions from denitrification increases exponentially when the WFPS exceeds the texture related threshold value and levels off as the soil approaches saturation. This suggests that there would be no decrease in N2O production at extremely high WFPS in Daycent? Which is missleading – please describe the whole relationship, also the decrease at high WFPS; (see Parton et al 2001)

*Answer 6: In Daycent, denitrification occurs in the interval 55%< WFPS <90% and increases exponentially but becomes static when soil approaches saturation (i.e. WFPS>90%). In addition, the portion lost as N2O decreases as WFPS increases – see answer to comment 2.*

**Changes in manuscript 6:** *This information has been added in Line 140-143.*

**Referee comment 7:** Table 1: The order of the paragraphs describing the sites should be consistent with the site order in the tables.

**Answer 7:** *Thank you, this was indeed inconsistent. We now adjusted the order of the table so that it is consistent with the paragraphs.*

**Changes in manuscript 7:** *The revised order can be found in Table 1 on page 8.*

**Referee comment 8:** Table 2: Soil pools: Units are incomplete: g C per what, g N per what? When I look at the C-N ratios in Table 2, it becomes obvious that something must be wrong here. In Michigan (obs) this is rather low (8.6), however in Nebraska a Soil C/N ratio of 1.2 – this is impossible. Please check your values for C and N pools thoroughly. If really the values as written were used in the modeling, the respective runs must be redone with corrected values.

**Answer 8:** *Thank you for pointing to this.*

- *The units were indeed incomplete and are now extended to g C kg$^{-1}$ dry soil for carbon and g N kg$^{-1}$ soil for nitrogen.*

- *There are indeed some mistakes in the reported values. All values have been double-checked, and the following corrections have been made for the site in Nebraska: (a) the soil C pool values of the observed data are corrected from 1762.2 to 17562.2; (b) erroneously reported simulated data from Colorado for the simulated soil pools of Nebraska are corrected and (c) the pool sizes referred to 0-100 cm soil depth rather then 0-150 cm soil depth. The correctly reported simulated soil C and soil N pools in Nebraska are 8769.7 soil C (g C kg$^{-1}$ soil) and 717.9 soil N (g N kg$^{-1}$ soil). The correct values were used for the model runs;*

*hence, no new runs are required.*

- *We also double checked the pool sizes and C/N ratio for Michigan, which were found to be correct. The C/N ratio of the experimental site in Michigan seems to be characterized by low values, see for example the summary table of the experimental site reported by Crum, J. R. and H. P. Collins (1995).*

***Changes in manuscript 8:*** *The corrected pool sizes and extended units can be found in Table 2 on page 9.*

**Referee comment 9:** L 192: all is applied at sowing - So does this mean exactly at sowing date? (-> farmers would often do it rather 1-2 weeks later.)

***Answer 9:*** *Currently, in the default setting of LPJmL, (part of) the fertilizers are indeed applied exactly at sowing. This is a simplification due to the fact that there is no spatially-explicit dataset available on fertilizer application dates with global coverage. However, fertilizer application at sowing or before sowing (e.g. during seed bed preparation) is not an uncommon practice, because combining agronomic operations is in some cases operationally advantageous for farmers.*

***Changes in manuscript 9:*** *We specified that fertilizers have been applied on the sowing date in Line 205.*

**Referee comment 10:** L 195: 0.7 for maize, what about the other crops?

***Answer 10:*** *Irrigation events occur when the fractional soil moisture of the water holding capacity (unitless) is below an irrigation threshold value of 0.7 for maize. The irrigation threshold value indeed depends on the crop type in LPJmL. C4 crops (e.g. maize) have a threshold value of 0.7 and C3 crops (e.g wheat and soybean) have a*

*threshold value of either 0.8 or 0.9, depending on the annual amount of precipitation. For a detailed description of irrigation schemes in LPJmL, we refer to Jägermeyr et al. (2015). In our analyses, we only focused on maize and therefore only reported the irrigation threshold value for maize. However, as some of the sites were cropped in a maize-wheat (Boigneville) or maize-wheat-soybean (Michigan) rotation, we now added the irrigation threshold values for wheat and soybean.*

***Changes in manuscript 10:*** *The reported irrigation threshold values for wheat and soybean can be found in Line 207-209.*

**Referee comment 11:** How do you explain the large differences in soil N between observed and simulated values (Table 2) in Michigan and Nebraska? The given depths are consistent between obs and sim, right? Overall, the method section is clearly written and comprehensive enough to allow the reader to thoroughly understand the modeling experiment.

***Answer 11:*** *Thank you for pointing to this difference and the need for an explanation. There are indeed large differences between the observed and simulated values. The differences between the observed and simulated pool sizes for Nebraska are partly due to the erroneously reported values (please, refer to the answer to comment 6). After correcting the pool sizes for Nebraska, the differences decreased, but are still relatively large compared to the reported pool sizes of the other experimental sites. The reason for these large differences is unclear. However, previous analyses have often shown poor agreement between measured and modeled soil mineral N values (see for example Del Grosso et al., 2008).*

***Changes in manuscript 11:*** *This information has been added in Line 439-442.*

**Referee comment 12:** L276: "We analyzed the experimental site in Nebraska". Here I was quite surprised that the analyses of soil moisture was only performed at one site. I would add quite early, when you first mention the soil moisture analyses in the Intro or beginning of methods the specification "at one selected site".

*Answer 12: Thank you for your suggestion. To clarify that the analysis of soil moisture is performed at one site, we modified the following sentence in the introduction: "Because of the importance of soil moisture for N2O emissions, we test the accuracy of the simulated soil moisture dynamics and its effects on N2O emissions against observations at one selected site in Nebraska, which was the only site with sufficient soil moisture data available."*

*Changes in manuscript 12: The adjusted sentence can be found in Line 57-60.*

**Referee comment 13:** L 298: "The significance of r corresponds to the tests, null hypothesis: r=0." The sentence as it is now, makes no sense since it is unclear to which test it refers. Please clarify.

*Answer 13: Here we calculated the significance of association between the measured and the simulated values through hypothesis testing, using the correlation t-test. The correlation coefficient is tested for significance to confirm if a real relationship exists between the measured and simulated values.*

*Changes in manuscript 13: We added the following information in Line 310-313: "We additionally calculated the significance of association between the measured and the simulated values through hypothesis testing, using the Student's t-test, indicating significance levels as "n.s." for p >=0.05, ∗ for p<0.05,∗ ∗ for p<0.01 for all r values.*

**Referee comment 14:** L 315: In this paragraph, it would help the reader if you first

briefly point out what tillage effect the observed data show, and only after that present the model results; then the reader has a chance to compare without again looking to the plot. As it is now, it is a bit cumbersome to read. I would strongly urge to order: Observed results, then model results, then details.

*Answer 14: We followed the suggestion and changed the order of the paragraph to: observed results, models results and then the details of the results.*

*Changes in manuscript 14: The revised paragraph can be found in Line 331ff*

**Referee comment 15:** L 336 Also in this paragraph sentences, I think for each it makes it easier to put first observed results (which I think shout be reference), then model results.

*Answer 15: We followed the suggestion and changed the order of the paragraph to: observed results, models results and then the details of the results.*

*Changes in manuscript 15: The revised paragraph can be found in Line 351ff*

**Referee comment 16:** Fig 2: Is this the difference between no-tillage and tillage, N2Onotill – N2Otill? You could add this to the caption.

*Answer 16: The effect of no-tillage (Fig. 2, Fig. 3 and Fig. 6) is indeed expressed as the difference between N2O emitted from no tillage and N2O emitted from tillage. This information is now added to the caption of Fig. 2, 3 and 6.*

*Changes in manuscript 16: The captions are adjusted in Fig. 2, Fig. 3 and Fig. 6.*

**Referee comment 17:** L 316: LPJmL.G.Orig showed an increase in emissions with

no-tillage (Fig. 2 A)

*Answer 17:* *We assume that you are asking to rephrase this sentence. We changed it to: "In response to no-tillage, LPJmL.G.Orig showed an increase in N2O emissions by 59.5% (Fig. 2A), and 22.4% in LPJmL.D.Orig. (Fig. 2B)."*

*Changes in manuscript 17:* *The rephrased sentence can be found in Line 335.*

**Referee comment 18:** *L 326: use "more detailed" instead of different*

*Answer 18:* *Thank you. We changed "different" into "more detailed".*

*Changes in manuscript 18:* *The change can be found in Line 341.*

**Referee comment 19:** L353: You give values for LPJmL only on no-tillage, but for DayCent tillage and notillage?

*Answer 19:* *We indeed missed to report the soil moisture values simulated by LPJmL for tillage. We now revised the sentence to report those values as well: "The soil moisture (WFPS) simulated by LPJmL.D.Orig in Nebraska, is high compared to the observed values for no-tillage (RMSD= 0.24 (unitless), r= 0.28) (Fig. 4) and tillage (RMSD= 0.21 (unitless), r= 0.10)."*

*Changes in manuscript 19:* *The revised sentence can be found in Line 369.*

**Referee comment 20:** Fig 4: You use RMSD throughout the manuscript, but switch to RMSE here: Probably this is a typo so you meant RMSD and referred to WFPS as a fraction? There is no need to extend the y axes from 0 to 1, which results in half of the area without information - instead the plot could be a bit larger.

[Figure]

*Answer 20:* *Thank you for pointing to these issues. We assume that you refer to Table 4 instead of Fig 4. We corrected "RMSE" to "RMSD" in table 4. For clarification, we now refer to the soil moisture as a fraction of the water filled pore space (WFPS) in the caption of table 4 and Fig.4. The y axes in Fig. 4 is now also narrowed ranging from 0.2 to 0.8 instead of from 0 to 1 to reduce the area without information.*

*Changes in manuscript 20:* *The adjusted Table 4 and Fig 4 can be found on page 17 and page 19 respectively.*

**Referee comment 21:** How much differ the PTFs used in Daycent and LPJml?

*Answer 21:* *In both models, the PTF is used to calculate soil hydraulic properties from soil texture. The PTF used in LPJmL is based on Saxton et al., (2006) whereas the PTF in DayCent is based on Saxton et al., (1986). The PTF of Saxton et al., (1986) was updated in Saxton et al., (2006) with new equations derived from a larger soil database using, next to commonly available variables of soil texture, also organic matter (see also Saxton et al., 2006). In LPJmL, this PTF was included in the model in order to dynamically simulate hydraulic parameters, i.e. the PTF allows for a dynamic effect of SOM on soil hydraulic properties, so that it is capable of representing changes in bulk density after tillage (for a detailed description we refer to Lutz et al., 2019).*
*In Figure 4 of the manuscript, the differences in field capacities and wilting points are due to the different PTFs. In the figure, the PTF used by LPJmL.D.Orig results in the simulation of a higher field capacity and wilting point (in grey) compared to the PTF used in DayCent (in blue).*

*Changes in manuscript 21:* *We added the following information in Line 379: "The PTFs used by both models to calculate soil hydraulic properties (e.g. FC and WP) that influence water dynamics do not fully account for the influence of soil structure which*

*likely contributes to model errors (Fatichi et al. 2020).*"

**Referee comment 22:** L 433: Under completely anaerobic conditions, no more N2O will be emitted, instead there occurs complete reduction to N2. An exponential increase with soil moistures at very high moisture levels does not make sense.

*Answer 22: We agree that under completely anaerobic conditions, denitrification does not lead to N2O emissions but rather the production of N2. In LPJmL, the N2O emissions from denitrification increase exponentially after the WFPS reaches $\sim80\%$, as described in Line 107-110. As N2 emissions are calculated as a fixed fraction of N2O, increases in N2O emissions also lead to an increase in N2 emissions. Although, in complete anoxic conditions, N2O is fully reduced to N2, this part of the process is not included in LPJmL. However, very high WFPS conditions rarely occur in the model for a long period of time and also not at the experimental sites used for this study (see also figure 4). Anoxia for a longer period of time mainly occurs in environments such as wetlands, paddy soils, areas with high water tables, organic soils and heavy textured soil (Inglett et al., 2005). The experimental sites are not located in such environments. Hence, complete anoxia conditions are typically not reached in our study here and the failure of the model to account for the effect that all denitrified N is emitted as N2 rather than as N2O likely has minimal relevance for our results. However, we have added this point to the discussion as warranted future improvement.*

*Changes in manuscript 22: This information has been added in Line 462-466.*

**Referee comment 23:** Fig A1: Adding r values for each run would help to get insights about the degree of association. The mean and distribution give only a very limited picture.

*Answer 23: We choose to exclude the r values in the figures to avoid an overload on information in the graph. Instead,we included those values (along with the RMSD,*

*mean bias and standard deviation) in Table A1 on page 28.We now point to these in the caption of Figure A1.*

***Changes in manuscript 23:*** *We changed the caption of Table A1 to: "Performance Daycent and LPJmL over all sites and years. RMSD is the root mean square deviation (in g N $ha^{-1}d^{-1}$), r is the correlation coefficient (unitless), p is the significance of r (unitless), MB is the mean bias (unitless) and SD is the standard deviation (in g N $ha^{-1}d^{-1}$)".*

**Referee comment 24:** Technical corrections

***Answer 24:*** *Thank you for the technical corrections. We revised the manuscript accordingly.*

***Changes in manuscript 24:***

- *We revised the Methods and Results section to use past tense.*
- *We changed Line 70 to the proposed sentence suggested by the reviewer: "Four experimental sites with detailed information on management available were identified" .*
- *Table 2: we rounded consistently.*
- *We changed Line 290 to the proposed sentence suggested by the reviewer: "we focused on the top 0.2 m of the soil" .*
- *We revised Line 302 to: "Therefore, we calculated the difference between simulated and observed values using root mean squared deviation (RMSD in g N $ha^{-1}$ $d^{-1}$) of the different sites as in equation 9:"*
- *Fig 3: Has now capital letters for all sites.*

- *Fig 3: We now use n = 123 instead of n= 123. The numbers on top of the box plots represent the median values. This information is added to the caption of the figure. Moreover, the median values are now rounded consistently.*

**References**

Crum, J. R. and H. P. Collins. 1995. KBS Soils. *Kellogg Biological Station Long-term Ecological Research Special Publication*. Zenodo. http://doi.org/10.5281/zenodo.2581504.

Del Grosso, S.J., Halvorson, A.D. and Parton, W.J., 2008. Testing DAYCENT model simulations of corn yields and nitrous oxide emissions in irrigated tillage systems in Colorado. Journal of environmental quality, 37(4), pp.1383-1389.

Inglett, P. W., Reddy, K., and Cortanje, R. 2005. "Anaerobic Soils." In Encyclopedia of Soils in the Envinroment, eduted by D. Hillel. Academic Press, pp.72-78.
Jägermeyr, J., Gerten, D., Heinke, J., Schaphoff, S., Kummu, M., and Lucht, W.: Water savings potentials of irrigation systems: global simulation of processes and linkages, Hydrol. Earth Syst. Sci., 19, 3073–3091, https://doi.org/10.5194/hess-19-3073-2015, 2015.

Lutz, F., Herzfeld, T., Heinke, J., Rolinski, S., Schaphoff, S., von Bloh, W., Stoorvogel, J. J., and Müller, C.: Simulating the effect of tillage practices with the global ecosystem model LPJmL (version 5.0-tillage), Geosci. Model Dev., 12, 2419–2440, https://doi.org/10.5194/gmd-12-2419-2019, 2019

Parton,W., Mosier, A., Ojima, D., Valentine, D., Schimel, D.,Weier, K., and Kulmala, A. E.: Generalized model for $N_2$and $N_2O$ production from nitrification and denitrification,
Global Biogeochem. Cycles, 10, 401–412, https://doi.org/10.1029/96GB01455, 1996.

Parton, W., Holland, E., Del Grosso, S., Hartman, M., Martin, R., Mosier, A., Ojima, D., and Schimel, D.: Generalized model for NOx and N2O emissions from soils, Journal of Geophysical Research: Atmospheres, 106, 17 403–17 419, https://doi.org/10.1029/2001JD900101, 2001.

Saxton, K., Rawls, W., Romberger, J., and Papendick, R.: Estimating generalized soil-water characteristics from texture, Soil Sci. Soc. Am. J., 50, 1031–1036, https://doi.org/10.2136/sssaj1986.03615995005000040039x, 1986.

Saxton, K. E. and Rawls, W. J.: Soil Water Characteristic Estimates by Texture and Organic Matter
for Hydrologic Solutions, Soil Sci. Soc. Am. J., 70, 1569–1577, https://doi.org/10.2136/sssaj2005.0117, 2006.

---

## Author Comment (AC2) · 15 Jun 2020

**Dear Editor and Referees,**
Thank you for the thorough evaluation of our manuscript and the very helpful and detailed feedback. This is much appreciated. We were able to address all of the reviewers' points, which helped to improve the scientific rigor and presentation of our work in this paper.

In this letter we list the referees' comments, each point followed by our responses, and

the changes in the manuscript.

The responses and subsequent modifications to the manuscript have been derived in consultation with all co-authors.

Best regards,
Femke Lutz

**Referee # RC2**
Lutz and co-authors validated a model that estimates soil N2O emissions in tillage and not tillage agriculture against field experiments. They report that (1) the model performance is improved by using including site-specific land use information as a model input instead of global model estimates and that (2) the model performance bias (overestimation of emissions) is reduced by a better parametrization of hydrological processes (to avoid an overestimation of soil moisture).
This is a well-structured manuscript that makes important contributions to the incremental improvement of the LPJmL5.0-tillage model. The manuscript is well structured and easy to read. Overall, I find the author work convincing and have only minor comments:

***Thank you for the positive general assessment.***

**Referee comment 1:** I recommend removing the grey background and grid form the plots to improve the figures readability.

*Answer 1: Thank you for your comment. We removed the grey background and grid to improve the readability of the figures as suggested.*

***Changes in manuscript 1:*** *The improved figures can be found throughout the entire manuscript.*

**Referee comment 2:** General discussion and conclusion sections are almost of the same length and largely redundant.

***Answer 2:*** *We agree that there is redundant information in those sections. Therefore, we shortened the conclusions by focusing on the main objectives of the work.*

***Changes in manuscript 2:*** *The modified version of the conclusion can be found in Line 475ff.*